# Evolving scattering networks for engineering disorder

Sunkyu Yu ● ✉

Network science provides a powerful tool for unraveling the complexities of social, technological and biological systems. Constructing networks using wave phenomena is also of great interest in devising advanced hardware for machine learning, as shown in optical neural networks. Although most wave-based networks have employed static network models, the impact of evolving models in network science provides strong motivation to apply dynamical network modeling to wave physics. Here the concept of evolving scattering networks for scattering phenomena is developed. The network is defined by links, node degrees and their evolution processes modeling multi-particle interferences, which directly determine scattering from disordered materials. I demonstrate the concept by examining network-based material classification, microstructure screening and preferential attachment in evolutions, which are applied to stealthy hyperuniformity. The results enable independent control of scattering from different length scales, revealing superdense material phases in short-range order. The proposed concept provides a bridge between wave physics and network science to resolve multiscale material complexities and open-system material design.

Evolving network models—the models that characterize the mechanisms of time-varying networks—have stimulated substantial advances in network science and related disciplines. One of the most important achievements in this field is the discovery of scale-free networks using the Barabási–Albert model[1,2], which describes the evolution process with stochastic network growth and the preferential attachment of new nodes to more connected ones. The model has been extended to other evolution processes to describe different forms of dynamics, such as the node fitness[3], accelerated growth[4] and aging models[5,6]. The pathway towards the resulting network of a given evolution process is not unique in general, as shown in scale-free networks developed by the deterministic process[7] or static models[8]. However, finding an underlying evolution process of a class of complex networks unveils their hidden traits and topologies, as demonstrated in the unique features of scale-free networks distinct from those of Erdős–Rényi random networks, such as power-law degree distributions[1], robustness to random failure[9], enhanced controllability[10] and ultrasmall-world

properties[11]. Recently, there has been a surge of interest in devising evolution processes for machine learning to impose more flexibility on neural networks for artificial general intelligence[12].

The use of network science is widespread throughout physics, as shown in quantum graph theory[13–16] and the network modeling of material states[3], potential landscapes[8] and interacting quantum processors[10]. In addition to understanding physics in material or structural networks, realizing networks defined by wave–matter interactions has received considerable attention associated with recent efforts to implement wave neural networks[17–25]. Such a wave network is composed of a set of wave 'nodes'—wave behaviors inside a unit element such as waveguides[19,26], resonators[21,27] and scatterers[20,28,29]—where wave interactions between the nodes determine wave 'links'. From this perspective, important achievements have brought a wider set of wave-based network structures, such as multiport interferometers for universal linear optics[26,30], diffractive multilayers for artificial neural networks[20] and engineered disorder for wave manipulations[29,31,32].

Intelligent Wave Systems Laboratory, Department of Electrical and Computer Engineering, Seoul National University, Seoul, Korea.
✉e-mail: sunkyu.yu@snu.ac.kr

Notably, quantum graphs provide an analytical tool to fully describe the interactions between wave nodes[13–16]. However, extracting the kernel of a wave network requires platform-specific simplification, as shown in the reflectionless and single-channel assumptions for the network modeling of guided[19,27,33] and diffractive[20] systems. In this context, the efficient network modeling of scattering phenomena with complex interferences from multiple particles is still an open question.

In terms of network classes, previous studies on wave networks have utilized static network models, which have a fixed number of wave nodes and use material perturbation to control the network structure. Although some evolutionary algorithms[34,35] have provided numerical tools for structural optimization, these methods lack the underlying concepts of evolving network models, such as the evolution process defined by the time-varying network topologies of the designed structures. Therefore, fundamental issues for a deeper understanding of wave networks—preferential attachment, evolving degree distributions and the impact of evolving models on engineering wave–matter interactions—remain open questions. When considering the revolutionary success of evolving models in network science, designing network-based evolution processes for wave–matter interactions will provide an insight into complex materials and artificial neural networks in photonics, acoustics and quantum graphs.

Here I propose the concept of evolving scattering networks—open-system wave-network models with a dynamically changing number of particles inside a system—which provides a tool for multiscale material design with target scattering responses. To offer a bridge between network science and wave physics, I define nodes, weighted links, degree distributions and evolution processes based on scattering theory. The suggested network model characterizes wave scattering as the network depending on the length scale of interest, which is suitable for designing materials with unique length-scale-dependent natures. As a representative example, I develop the evolution process towards stealthy hyperuniformity (SHU), which indicates the structural characteristics from the bounded suppression of density fluctuations at long-range scales[32,36–39] and has been studied in numerous natural or engineered systems, such as cosmological models[40], avian photoreceptors[41], amorphous silica[42], space partitioning[43], prime numbers[44], band-gap materials[31,45,46] and sunlight absorbers[47]. The proposed evolution process enables the network-based classification of material phases and the screening of existing materials with SHU states. By realizing the preferential attachment for the SHU evolution, I demonstrate the engineering of the degree distribution of scattering networks, achieving unconventional material states such as effectively denser or sparser particle distributions for short-range order while preserving long-range order of crystals or Poisson processes. The approach provides the network-based interpretation of wave phenomena, extending the candidate platforms for wave neural networks.

## Results

### Evolving scattering networks

To develop an evolving network model for waves, I start with the relationship between wave scattering and spatial ordering, widely employed in crystallography[48], statistical mechanics[49] and disordered photonics[32]. Consider a material composed of $n$ identical point particles located inside the finite-size spatial domain $\Omega$ (Fig. 1a), where the $j$th particle is at position $\mathbf{r}_j \in \Omega$ ($j = 1, 2, \ldots, n$). Under some assumptions (Methods), the scattering intensity measured at position $\mathbf{R}$ is $I_n(\mathbf{k}; \mathbf{R}) = nI_1(\mathbf{k}; \mathbf{R})S_n(\mathbf{k})$ (Supplementary Note 1), where $\mathbf{k} = \mathbf{k}_S - \mathbf{k}_I$ is the wavevector shift between incident ($\mathbf{k}_I$) and scattering ($\mathbf{k}_S$) waves ($|\mathbf{k}_I| = |\mathbf{k}_S| = k_o$ and $|\mathbf{k}| = k$), $I_1$ is the scattering intensity from a single particle and $S_n(\mathbf{k})$ is the structure factor:

$$S_n(\mathbf{k}) = \frac{1}{n}\left|\sum_{j=1}^{n} e^{-i\mathbf{k}\cdot\mathbf{r}_j}\right|^2, \tag{1}$$

which satisfies $S_1(\mathbf{k}) = 1$ and $S_n(\mathbf{k}) = S_n(-\mathbf{k})$. In this simplified model, wave scattering is governed by $S_n(\mathbf{k})$, which represents the spatial ordering of a material. Note that there is a one-to-many relationship between the structure factor and a set of particle positions[50], as shown in the loss of phase information due to the modulus in equation (1). Therefore, it is natural to classify materials with their scattering responses[32], for example, leading to the forward (Fig. 1b), backward (Fig. 1c) and zero (Fig. 1d) scattering states for the impulse-type spatial order $S_n(\mathbf{k}) = \delta(\mathbf{k} - \mathbf{k}')$, where $\delta(\mathbf{k})$ is the Dirac delta function (Methods and Supplementary Note 2). Such a classification underlines the necessity of engineering each regime of $S_n(\mathbf{k})$ independently. For this purpose, I develop an evolution model for scattering phenomena (see Methods for criteria of the evolution model).

Because $S_n(\mathbf{k})$ of a material determines wave scattering, realizing an evolution of $S_n(\mathbf{k})$ corresponds to designing the evolution of an $n$-particle material and its scattering responses. Similar to the growth in evolving networks[1,2,51,52], suppose an evolution model that describes the inclusion of a new particle to an existing $n$-particle material (Fig. 1e) according to a specific evolution process. By examining the evolution of the structure factor (Methods and Supplementary Note 3), I develop the network modeling of material and its scattering responses by defining the 'scattering network' composed of scatterer nodes, which are connected to each other through interference links: $\cos[\mathbf{k} \cdot (\mathbf{r}_p - \mathbf{r}_q)]$ between the $p$th and $q$th nodes (Fig. 1f). Because a material generally provides diverse $\mathbf{k}$ components[32,50], it is necessary to estimate the collective contribution of $S_n(\mathbf{k})$ in the reciprocal space to characterize the overall scattering responses of a given material. Therefore, a more rigorous definition of the link weight between the $p$th and $q$th nodes should be:

$$w_{p,q}{}^{\mathbf{K}} = \frac{1}{V_{\mathbf{K}}} \int_{\mathbf{K}} \cos\left[\mathbf{k} \cdot (\mathbf{r}_p - \mathbf{r}_q)\right] d\mathbf{k}, \tag{2}$$

where $\mathbf{K}$ is the region of interest in the reciprocal space and $V_{\mathbf{K}}$ is the volume of the space $\mathbf{K}$.

In terms of the network structure, the proposed scattering network is fully connected with undirected and weighted links $w_{p,q}{}^{\mathbf{K}}$ because the interference originating from a particle affects all the existing particles reciprocally and differently. Another feature of the scattering network is that $w_{p,q}{}^{\mathbf{K}}$ does not vary monotonically with the spatial distance due to the $\mathbf{k}$-dependent periodicity of $\cos[\mathbf{k} \cdot (\mathbf{r}_p - \mathbf{r}_q)]$ (Fig. 1f), in sharp contrast to other macroscopic or microscopic real-space networks, such as airline systems[2] or potential energy landscapes[8]. Such uniqueness highlights the necessity of wave-specific network modeling and evolution processes to engineer the material network topology and the consequent wave scattering.

From equation (2), the node degree—the connectivity of the $p$th node to the entire network[52]—becomes $w_p{}^{\mathbf{K}} = \sum_{q \neq p} w_{p,q}{}^{\mathbf{K}}$, which allows for the network-based interpretation of scattering:

$$\langle S_n \rangle_{\mathbf{K}} = \frac{1}{V_{\mathbf{K}}} \int_{\mathbf{K}} S_n(\mathbf{k}) d\mathbf{k} = 1 + \frac{1}{n} \sum_{p=1}^{n} w_p{}^{\mathbf{K}}. \tag{3}$$

Equation (3) shows that the scattering averaged in the $\mathbf{K}$ space $\langle S_n \rangle_{\mathbf{K}}$ is determined by the average node degrees. This result inspires the engineering of each node degree $w_p{}^{\mathbf{K}}$ while preserving $\sum w_p{}^{\mathbf{K}}$ through the designed evolution, similar to finding hub nodes in network science[1,51,52].

The features of an evolving network are determined by its evolution process[52]. For example, in the Barabási–Albert model[1,2], the preferential attachment of a new node to the existing nodes with higher node degrees—the rich get richer rule—results in the power-law scaling in node linkages, constructing scale-free networks. In evolving scattering networks, my goal is to engineer the network structure and the corresponding wave scattering by devising the proper rule to determine

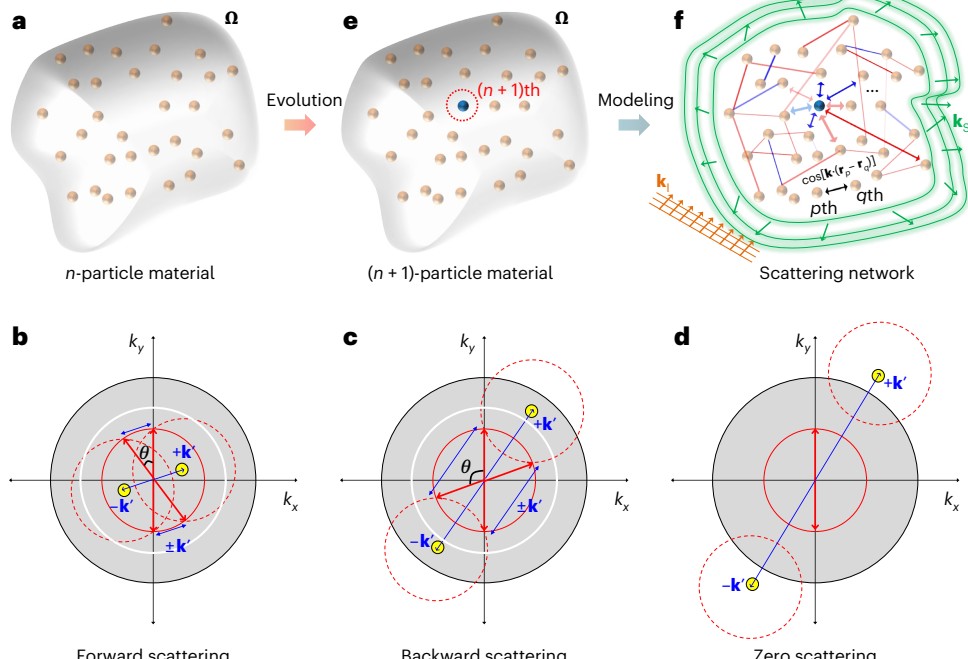

**Fig. 1 | Evolution of scattering networks. a**, A material composed of $n$ identical point particles in the spatial domain $\Omega$. **b–d**, Three different states of impulse scattering responses depending on the spatial ordering of materials: forward ($|\mathbf{k}'| < \sqrt{2}k_o$ for $0 \le \theta \le \pi/2$ or $3\pi/2 \le \theta \le 2\pi$) (**b**), backward ($\sqrt{2}k_o \le |\mathbf{k}'| < 2k_o$ for $\pi/2 \le \theta \le 3\pi/2$) (**c**) and zero ($|\mathbf{k}'| \ge 2k_o$) (**d**) scattering states. Yellow points denote the impulse $S_n(\mathbf{k}) = \delta(\mathbf{k} - \mathbf{k}')$. Red arrows represent the allowed incident and scattering wavevectors connected through the impulse scattering response $\pm\mathbf{k}'$. Red, white and black solid circles have the radii of $k_o$, $\sqrt{2}k_o$ and $2k_o$, respectively, where the red one represents the light cone. Red dashed circles are the shifted light cones due to the scattering events described by $S_n(\mathbf{k}) = \delta(\mathbf{k} - \mathbf{k}')$. **e**, An $(n+1)$-particle material evolved from the material in **a**, by adding the $(n+1)$th

point particle (blue sphere). **f**, Network modeling of wave scattering from a material with scatterer nodes and $\mathbf{k}$-dependent links. Orange and green arrows denote incident and scattering waves, respectively. Red and blue solid lines represent the positive and negative signs of existing link weights defined by equation (2), respectively. Red and blue arrows also represent the positive and negative signs of newly included link weights after adding the $(n+1)$th particle, respectively. Only the links with large values of $|w_{p,q}^{\mathbf{K}}|$ are assumed to be plotted because a scattering network is fully connected. The black arrow describes the $\mathbf{k}$-impulse component $\cos[\mathbf{k} \cdot (\mathbf{r}_p - \mathbf{r}_q)]$ of the link weight between the $p$th and $q$th particles. The transparency of the solid lines and arrows denotes the magnitude of the weights.

the position of the $(n+1)$th particle in an existing $n$-particle material. I develop the evolution process for scattering networks (Methods), which is defined by the $\mathbf{K}$-dependent cost function $\rho_n^{\mathbf{K}}(\mathbf{r})$ for the $n$th evolution:

$$\rho_n^{\mathbf{K}}(\mathbf{r}) = \frac{1}{n} \sum_{p=1}^{n} \Pi(w_p^{\mathbf{K}}) \left[ \frac{1}{V_{\mathbf{K}}} \int_{\mathbf{K}} \cos\left[\mathbf{k} \cdot (\mathbf{r}_p - \mathbf{r})\right] d\mathbf{k} \right], \quad (4)$$

where $\Pi(w_p^{\mathbf{K}})$ is the preference function, which characterizes the preferential attachment to the $p$th node. The minimum of $\rho_n^{\mathbf{K}}(\mathbf{r})$ represents the best position for the $(n+1)$th particle. Because the term in the square brackets is the link weight between the $p$th node and the $(n+1)$th node at $\mathbf{r}$, $\rho_n^{\mathbf{K}}(\mathbf{r})$ directly denotes the node degree of the $(n+1)$th node when $\Pi(w_p^{\mathbf{K}}) = 1$, leading to the evolving change of $\langle S_n \rangle_{\mathbf{K}}$ in equation (3). The control of $\Pi(w_p^{\mathbf{K}})$ differentiates the importance of each existing node in altering $\langle S_n \rangle_{\mathbf{K}}$, eventually imposing the 'preference' on each node during the evolution process. By changing $\mathbf{K}$, can also manipulate wave scattering across different scattering states in Fig. 1b–d (see Supplementary Algorithm 1 for pseudo-code form of the evolution process). I also generalize the evolving network concept to inhomogeneous materials in Methods and Supplementary Note 4.

Figure 2 describes the evolution process, designing the material inside the real space $\Omega$ to achieve $\langle S_n \rangle_{\mathbf{K}} \to 0$ (see Methods for the preconditions and parameters of the evolution process). I employ the Monte Carlo method to sample both real ($\mathbf{r} \in \Omega$) and reciprocal ($\mathbf{k} \in \mathbf{K}$) spaces (Fig. 2a,b and Supplementary Note 5), which allows for the statistically homogeneous and isotropic evaluation of both spaces.

The evolution of the cost function $\rho_n^{\mathbf{K}}(\mathbf{r})$, which illustrates the probability map of placing a new particle, demonstrates that the evolution process consumes the finite real space $\Omega$ as the 'resource' to suppress wave scattering (Supplementary Notes 6 and 7 and Supplementary Video 1). Through the sequential finding of $\mathbf{r}_{n+1}$ from this process (Fig. 2c,d), the target evolution of $S_n(\mathbf{k})$ is then successfully achieved (Fig. 2e,f and Supplementary Note 7). The relationship between the random defects in the designed particle positions $\{\mathbf{r}_p\}$ and the following perturbation of $\langle S_n \rangle_{\mathbf{K}}$ is analyzed in Supplementary Note 8. The analysis shows that the particles with highly negative node degrees $w_p^{\mathbf{K}}$, which correspond to more important particles in realizing the SHU state, have stronger defect immunity at the weak defect regime. It is in sharp contrast to the fragile hub nodes in network science[2,52], demonstrating the uniqueness of the proposed scattering networks.

In the following sections, I examine the listed concepts of evolving scattering networks in a step-by-step manner with the examples of (1) classifying microstructures with a wave-network viewpoint, (2) screening microstructures using evolution processes and (3) controlling long-range and short-range order independently using preferential attachment.

## Material classification with degree distributions

I employ the concept of scattering networks to classify material microstructures. To focus on the criteria for classification, I assume the most straightforward form of the evolution process without preferential attachment as $\Pi(w_p^{\mathbf{K}}) = 1$, which imposes statistical homogeneity on all existing nodes. The evolution process leads to (Supplementary Note 9):

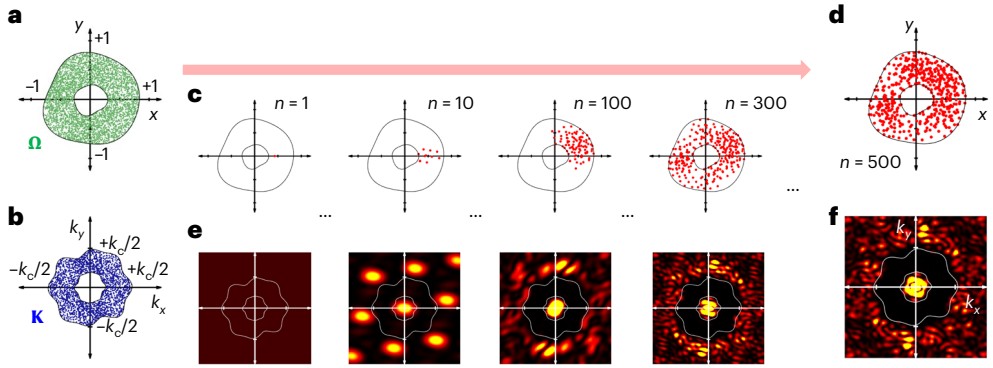

**Fig. 2 | Evolution process. a,b,** The Monte Carlo discretization of the real ($\Omega$; **a**) and reciprocal (**K**; **b**) spaces of interest. $k_c$ is the upper limit of the spatial frequency of interest for calculating $S_n(\mathbf{k})$, which is determined by the number of particles (Methods). **c,** The evolution of material in the real space. **d,** The resulting material with 500 particles. **e,** The evolution of the structure factor $S_n(\mathbf{k})$, corresponding to each material in **c. f,** $S_{500}(\mathbf{k})$ of the material in **d.** $\Pi(w_p{}^\mathbf{K}) = 1$ in this example. See Supplementary Video 2 for the evolutions of **c** and **e.**

$$\rho_n{}^\mathbf{K}(\mathbf{r}_{n+1} = \mathbf{r}_{\min}) = \min\left(\frac{n+1}{n}\langle S_{n+1}\rangle_\mathbf{K} - \langle S_n\rangle_\mathbf{K} - \frac{1}{n}\right), \qquad (5)$$

which has the thermodynamic limit of $\lim_{n\to\infty}\rho_n{}^\mathbf{K}(\mathbf{r}_{\min}) = \min$ $(\langle S_{n+1}\rangle_\mathbf{K} - \langle S_n\rangle_\mathbf{K})$ that results in the suppressed scattering in **K**.

Although various materials can be developed with different **K** (Fig. 2b), one of the most insightful examples obtained with equation (5) is hyperuniform materials[32,36,39], which require $S_n(\mathbf{k}) \to 0$ when $|\mathbf{k}| \to 0$ in the thermodynamic limit $n \to \infty$. Such suppression of infinite-wavelength density fluctuations has generalized the long-range order[32,39,43,53–55] of crystals, quasicrystals and correlated disorder, also revealing hidden order in photoreceptors[41] and jamming[56]. A stricter condition of SHU[32,37–39], requiring the bounded suppression of density fluctuations for the threshold $k_{th}$ as $S_n(\mathbf{k}) \approx 0$ for $|\mathbf{k}| < k_{th}$, has attracted considerable attention to elucidate complete bandgaps[31,46] (backscattering only in $S_n(\mathbf{k})$; Fig. 1c) and transparency[57] (zero scattering only in $S_n(\mathbf{k})$; Fig. 1d) in disordered materials.

To realize SHU materials with the evolution process, I define the reciprocal space for long-range order $\mathbf{K}_L = \{\mathbf{k} \mid k_{\min} \leq |\mathbf{k}| \leq k_{th}\}$, where $k_{\min}$ reflects numerically insuppressible scattering of near-infinite wavelengths due to the finite-range real space $\Omega$. I also define the reciprocal space for short-range order $\mathbf{K}_S = \{\mathbf{k} \mid k_{th} < |\mathbf{k}| \leq k_c\}$ to develop network quantities as another microstructural descriptor at shorter length scales. With this reciprocal-space design process, I revisit the comparison among the uncorrelated Poisson disorder (Fig. 3a), evolving SHU material (Fig. 3b) and square-lattice crystal having a similar SHU condition to that of the evolving material (Fig. 3c) to interpret the length-scale natures of each material state with the network concept (see Methods for parameters and Supplementary Note 10 for the crystals of different $n$). The SHU materials are obtained with the evolution by minimizing $\rho_n{}^{\mathbf{K}_L}(\mathbf{r})$, eventually suppressing $\langle S_n\rangle_{\mathbf{K}_L}$.

Although Fig. 3a–c shows a well-known structure factor of each material phase, Fig. 3d–i demonstrates that node degree distributions of evolving scattering networks operate as a useful tool for characterizing microstructures at the length scale of interest, bridging network analysis and material statistics. First, Poisson materials show Gaussian-like broad degree distributions around $w_p{}^{\mathbf{K}_L,\mathbf{K}_S} = 0$ at both long-range (Fig. 3d) and short-range (Fig. 3g) scales. In contrast, the crystal shows the opposite signs of the narrowband node degrees at long-range ($w_p{}^{\mathbf{K}_L} < 0$; Fig. 3f) and short-range ($w_p{}^{\mathbf{K}_S} > 0$; Fig. 3i) scales, which are determined by the first Bragg peaks (Supplementary Note 10). The network quantities clarify the uniqueness of SHU materials, exhibiting the crystal-like, narrowband negative degrees at the long-range scale

(Fig. 3e) and the Poisson-like, broad degree distribution at the short-range scale (Fig. 3h). Such network quantities can be described by the intuitive illustration of microstructures using node degrees (Fig. 3j–l), which clearly shows the contribution of each particle to scattering in terms of the strength and phase of interference. I also investigate the evolution of averaged scattering $\langle S_n\rangle_{\mathbf{K}_L}$ and $\langle S_n\rangle_{\mathbf{K}_S}$ for the Poisson and SHU processes (Supplementary Note 11), presenting the effect of the finite real space on the evolution and uniqueness of SHU materials in terms of particle density.

## Evolving material screening

Figure 3 describes the role of degree distributions in analyzing microstructures. However, the results in Fig. 3 do not show the potential of evolving scattering networks, as shown in the same features of evolving SHU materials (Fig. 3b) and conventional SHU materials[39], such as crystal-like long-range order and Poisson-like short-range order. Notably, the material design based on evolving scattering networks possesses open-system natures similar to dynamical additions of nodes and edges in graph neural networks[58], allowing for the alteration of matter (that is, an increasing particle number) and energy (that is, minimizing the cost function) inside the design domain. Therefore, the properties of evolving networks should be clarified with dynamical open systems[1,2,4–6,52]. As the first example, I investigate the evolution process applied to existing materials, especially focusing on the SHU evolution process to achieve the 'screening' of the microstructural properties of existing materials (see Methods for the comparison with traditional methods).

To examine the screening effect, I compare the scattering responses from different sequences of the Poisson process and the SHU process with non-preferential attachment (see Methods for parameters). While the 'SHU → Poisson' sequence (Fig. 4a–c) represents a simple combination of the material states in Fig. 3a,b, the 'Poisson → SHU' sequence (Fig. 4d–f) corresponds to the further growth of a Poisson material through the SHU evolution process using equation (5). I also compare different configurations by changing the allowed real spaces for each process: SHU mixing with the overlapped spaces (Fig. 4a,d), SHU core (Fig. 4b,e) and cladding (Fig. 4c,f) with the separated spaces. Despite the same number of particles for each configuration, all the results in Fig. 4a–f demonstrate the efficient suppression of long-range scattering using evolving scattering networks, as shown in the decrease of $w_p{}^{\mathbf{K}_L} > 0$ in Fig. 4d–f (reduced blue markers) compared with Fig. 4a–c. This efficient suppression, successfully 'screening' the microstructural property of Poisson materials, is also proved with the evolutions of averaged long-range scattering $\langle S_n\rangle_{\mathbf{K}_L}$ (Fig. 4g,h) for

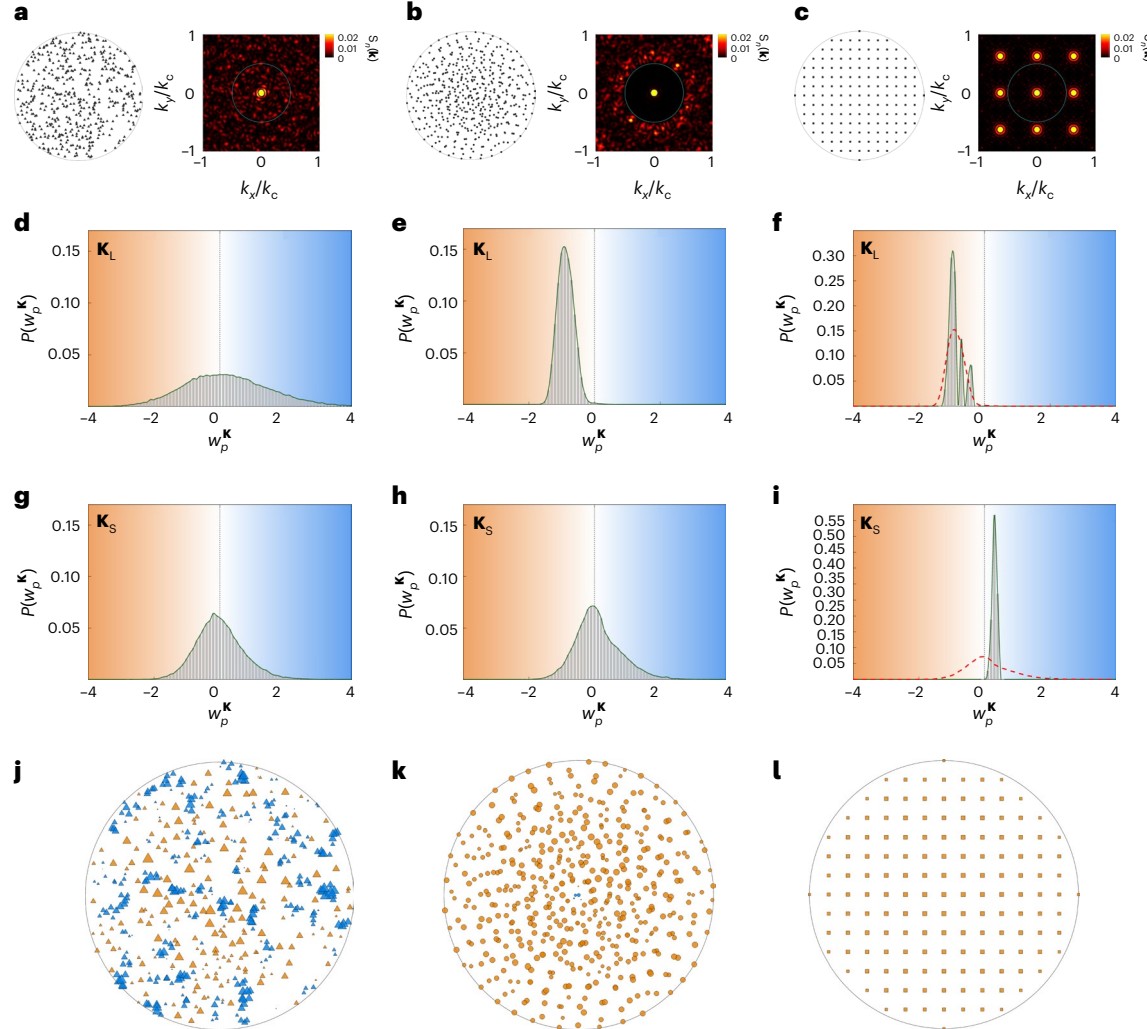

**Fig. 3 | Network-based classification of materials using degree distributions.** **a–l**, Three different material phases are analyzed: Poisson materials (**a,d,g,j**, triangles in **j**), SHU materials obtained from the evolution process (**b,e,h,k**, circles in **k**) and a square-lattice crystal with 149 particles (**c,f,i,l**, squares in **l**). **a–c**, Examples of each material phase and the resulting structure factor. **d–i**, Node degree distributions for the long-range scale with $\mathbf{K} = \mathbf{K}_L$ (**d–f**) and short-range scale with $\mathbf{K} = \mathbf{K}_S$ (**g–i**), where $P(w_p^{\mathbf{K}})$ represents the probability density distribution. The red and blue background colors represent negative (or suppressing) and positive (or enhancing) contributions to scattering, respectively. The red dashed lines in **f** and **i** are the results of **e** and **h**, respectively. When calculating degree distributions, a random ensemble of 100 realizations each with 500 particles is investigated each for Poisson and SHU material. **j–l**, Visualizations of materials with the node degrees for $\mathbf{K}_L$, where the orange and blue markers denote the negative and positive node degrees, respectively. The size of a marker represents the magnitude of the node degree $|w_p^{\mathbf{K}_L}|$. See Methods for detailed parameters.

an ensemble of realizations, showing much smaller values of $\langle S_{500}\rangle_{\mathbf{K}_L}$ in Fig. 4h than those of Fig. 4g (Supplementary Note 12). Because $\langle S_n \rangle_{\mathbf{K}}$ directly reflects the average node degrees in $\mathbf{K}$, Fig. 4g,h represents the dynamical evolutions of scattering network structures in terms of long-range and short-range order, which show the contrasting network structures between the simple SHU–Poisson combination (Fig. 4g) and the evolution-based screening (Fig. 4h), with positive and negative average node degrees in $\mathbf{K}_L$, respectively.

## Preferential attachment

Although Fig. 4 shows one of the interesting applications of evolving scattering networks, the employed evolution process maintains the non-preferential attachment with $\Pi(w_p^{\mathbf{K}}) = 1$. Historically, the critical impact of evolving networks has originated from preferential attachment, as demonstrated in the discovery of scale-free networks using the rich get richer rule[1,2]. Similarly, by engineering the preference function $\Pi(w_p^{\mathbf{K}})$, I can manipulate evolving scattering networks through the designed evolution rule, such as 'strong scatterers get stronger (or

weaker)', where the scattering strength of each particle is quantified by the node degree $w_p^{\mathbf{K}}$ from equation (3). Because the size and value of the array $\{w_p^{\mathbf{K}} \mid p = 1, 2, …, n\}$ changes during the evolution, the preferential attachment is a dynamical process, in sharp contrast to the static or generative methodologies with preassigned rules, such as the collective coordinate method[37,57,59] or its extension to molecular dynamics[38,60].

As an example, I examine the tangent hyperbolic preference function:

$$\Pi(w_p^{\mathbf{K}}) = \Pi_0 - \Pi_1 \tanh\left[\alpha\left(w_p^{\mathbf{K}} - w_c^{\mathbf{K}}\right)\right], \qquad (6)$$

where $\Pi_1 > 0$ and $\alpha$ determine the variation amplitude and slope of the function, $w_c^{\mathbf{K}}$ is the center degree and $\Pi_0$ is set to $\min[\Pi(w_p^{\mathbf{K}})] = 1$. I apply equation (6) to the SHU process, again trying to suppress the fluctuation in the long-range scale by setting $\mathbf{K} = \mathbf{K}_L$.

In Fig. 5, I examine opposite forms of preference in the evolution process (insets in Fig. 5b,e,h); weak scatterers get weaker ($\alpha > 0$; Fig. 5a–f) and strong scatters get weaker ($\alpha < 0$; Fig. 5g–i) in terms of the

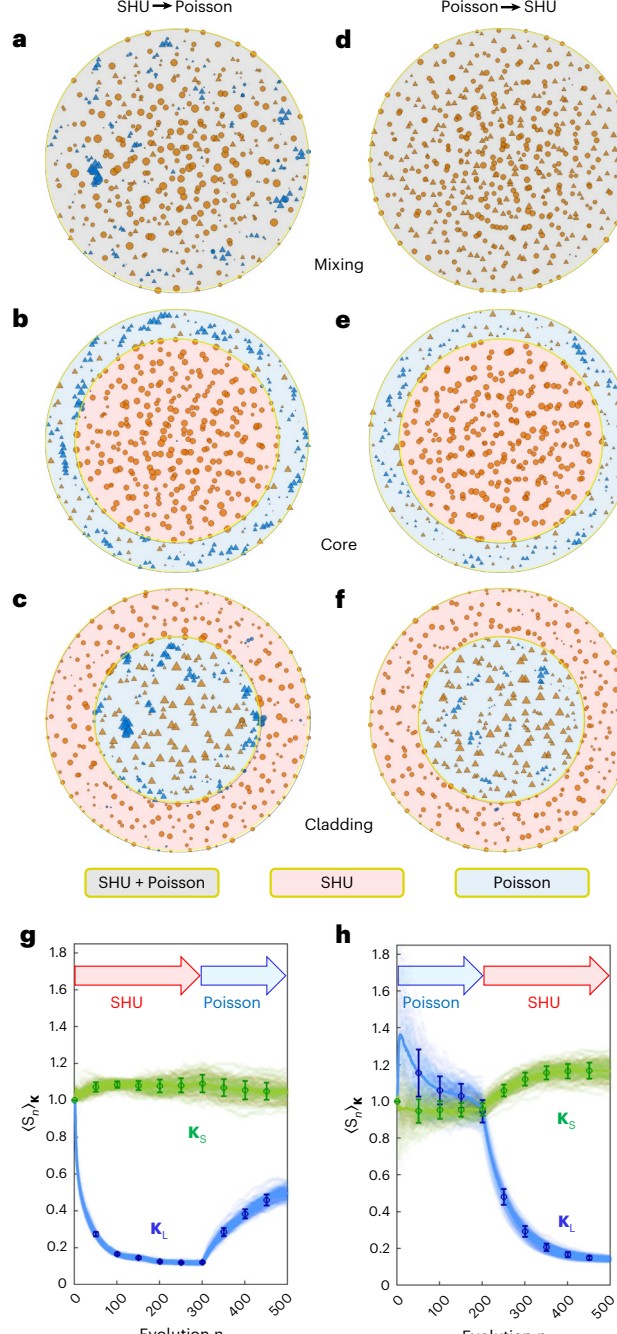

**Fig. 4 | Evolving SHU screening of Poisson materials. a–f**, Visualizations of materials with the node degrees for $K_L$, where the orange and blue markers denote the negative and positive node degrees, respectively: SHU → Poisson processes (**a–c**) and Poisson → SHU processes (**d–f**). The size of a marker represents the magnitude of the node degree $|w_p^{K_L}|$. Among $n = 500$ particles for each realization, SHU and Poisson processes cover 300 and 200 particles, respectively. The realizations in **a–f** are classified according to the allowed region of the real space for each process: SHU mixing configuration sharing the entire space for SHU and Poisson processes (gray shaded area) (**a,d**), SHU core configuration with an inner SHU region (red shaded area) and outer Poisson region (blue shaded area) (**b,e**) and SHU cladding configuration with an inner Poisson region (blue shaded area) and outer SHU region (red shaded area) (**c,f**). **g,h**, Evolutions of $\langle S_n \rangle_{K_L}$ (blue points) and $\langle S_n \rangle_{K_S}$ (green points) during the processes for the mixing configuration in **a** and **d**. A random ensemble of 100 realizations is investigated for both SHU → Poisson (**g**) and Poisson → SHU (**h**) sequences. In **g** and **h**, circles and error bars represent the mean value and one standard deviation of each ensemble of 100 realizations, respectively. See Methods for detailed parameters.

long-range scale $K_L$. The former (Fig. 5a–f) corresponds to preferential attachment, while the latter represents anti-preferential attachment. To examine scattering with network quantities, I separately calculate the degree distributions of long-range ($K = K_L$; Fig. 5b,e,h) and short-range ($K = K_S$; Fig. 5c,f,i) scales.

In contrast to the abstract modeling in network science, the finite real space $\Omega$ restricts the allowed position for including particles according to wave interferences (Supplementary Note 6), eventually limiting the possible range of scattering strength. Therefore, weak preference (Fig. 5a,b) leads to an almost similar result in $K_L$ to that of non-preferential attachment (Fig. 3e,k). Although strong preference derives a 'long tail' in $w_p^K < 0$ (large orange circles in Fig. 5d and red arrow in Fig. 5e), which corresponds to hub nodes in scale-free networks[1,2], the attempt to satisfy 'weak scatterers get weaker' in the finite $\Omega$ makes strong scatterers be stronger (large blue circles in Fig. 5d and black arrow in Fig. 5e). Similar observations can also be found in the scattering from the short-range scale $K_S$, showing marginally enhanced suppression of scattering in weak scatterers (red arrows in Fig. 5c,f) but with increased scattering in strong scatterers for enhanced preference (black arrow in Fig. 5f).

Another intriguing example is achieved with anti-preferential attachment (Fig. 5g–i). The attempt to suppress strong scatterers in the long-range scale maintains the criterion for SHU materials, resulting in a similar degree distribution to that of non-preferential attachment (Fig. 5h). However, as shown in the clustering of particles in Fig. 5g, anti-preferential attachment leads to the substantially enhanced scattering in the short-range scale $K_S$ (Fig. 5i), which can be understood as the side effect of the complete suppression of scattering in the long-range scale in the finite space $\Omega$. Based on the distinct results in Fig. 5, which strongly depend on the form of preferential attachment, we can engineer long-range and short-range scatterings independently, distinct from the conventional SHU state.

Similar to the finding of novel network topologies using evolving networks[1,2,52], Fig. 6a shows the impact of evolving scattering networks in exploring material phases. Compared with conventional SHU (phase I, green shaded region) and crystal (cross markers), weak-preferential (phase II) and anti-preferential (phase IV) attachments cover a substantially extended range of engineering short-range scattering while preserving the SHU condition with suppressed long-range scattering. On the other hand, strong preferential attachment (phase III) not only enables the gradual transition from the SHU to the near Poisson state but also achieves better short-range scattering over that of the Poisson material. These results demonstrate that evolving scattering networks with preferential attachment enable the discovery of the vast intermediate regime between order and uncorrelated disorder[32], also achieving unique scattering distinct from crystalline, Poisson and conventional SHU materials.

I also examine the evolutions of averaged long-range and short-range scatterings (Fig. 6b–d and Supplementary Note 13), showing two branch points in the material phase transition. The first branch point is the particle number $n_{B1} \approx 125$ (black dashed arrows in Fig. 6b–d), which gives the upper limit of the spatial frequency $k_c = 2\pi/d_c$ smaller than $k_{th}$ of the target SHU condition, where $d_c$ is the characteristic distance defined in Methods. When $n < n_{B1}$, the effect of the finite $\Omega$ is negligible, and thus all the SHU evolutions provide similar behaviors: gradual increases of short-range scattering $\langle S_n \rangle_{K_S}$ while achieving $\langle S_n \rangle_{K_L} \rightarrow 0$. When $n > n_{B1}$, the preferential attachment starts to govern material phases; preference (phases II and III) and anti-preference (phase IV) derive the suppressed and enhanced short-range scattering $\langle S_n \rangle_{K_S}$, respectively, while maintaining the SHU condition. When considering the scattering responses of the crystals in Fig. 6a (cross symbols), the branch point $n_{B1}$ leads to the separation of crystal-like SHU phases with preferential attachment (phases II and III) and non-crystal-like SHU phases with anti-preferential attachment (phase IV). The second branch point is the particle number $n_{B2} \approx 350$ (red dashed arrows

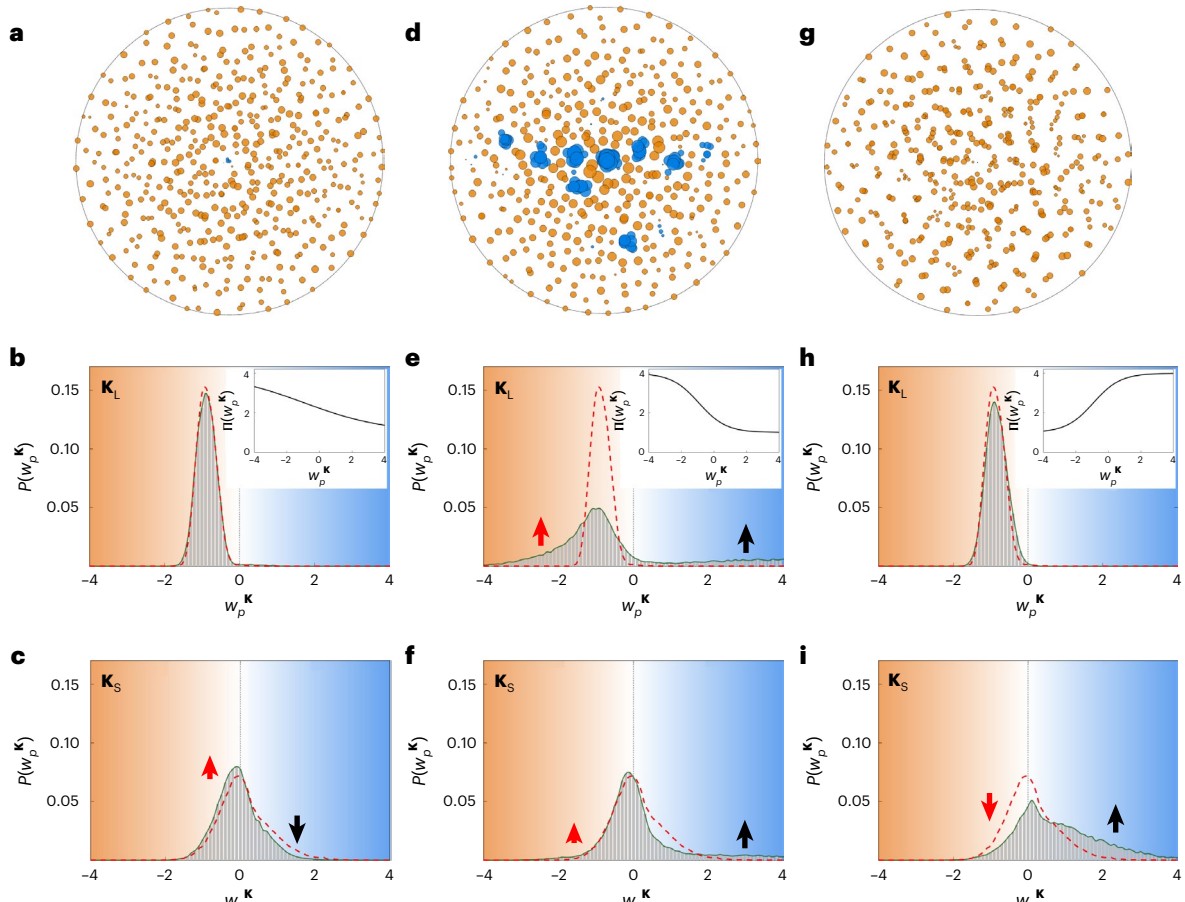

**Fig. 5 | Evolving scattering networks with preferential attachment. a–c**, Weak preference ($\alpha = 0.2$). **d–f**, Strong preference ($\alpha = 0.6$). **g–i**, Strong anti-preference ($\alpha = -0.6$). **a,d,g**, Visualizations of node degrees for the long-range scale $\mathbf{K}_L$, where the orange and blue markers denote the negative and positive node degrees, respectively. The size of a marker represents the magnitude of the node degree $|w_p{}^{\mathbf{K}_L}|$. **b,c,e,f,h,i**, Node degree distributions for the long-range scale $\mathbf{K}_L$ (**b,e,h**) and short-range scale $\mathbf{K}_S$ (**c,f,i**). Red dashed lines denote the results from

the SHU materials with non-preferential attachment (Fig. 3e,h). Red and black arrows denote the changes of $P(w_p{}^{\mathbf{K}})$ in the negative and positive ranges of $w_p{}^{\mathbf{K}}$, respectively, according to the introduction of preference or anti-preference. Insets in **b**, **e** and **h** show the preference function $\Pi(w_p{}^{\mathbf{K}})$. For all cases, $\Pi_0 = 2.5$, $\Pi_1 = 1.5$ and $n = 500$. $w_c{}^{\mathbf{K}} = -0.926$, which is the value of the peak in Fig. 3e. A random ensemble of 100 realizations is investigated for each preference function. All the other parameters are the same as those in Fig. 3b.

in Fig. 6b–d), which corresponds to the number of particles when the Bragg peaks are getting out of $\mathbf{K}_L$ (Supplementary Fig. 6b). Too strong preference exhausts the 'resource'—the candidate positions in the real space $\mathbf{\Omega}$ for suppressing long-range scattering—for the evolving process after this branch point ($n > n_{B2}$). The phases of preferential attachment (phases II and III) are then separated, realizing unconventional material states that support superior short-range scattering to Poisson materials (phase III).

The results in Figs. 3–6 are obtained from the assumption of identical point particles. I also examine the validity of the theory in another aspect by conducting the full-wave analysis (Supplementary Note 14). This numerical analysis reflects the effects of finite-size particles and multiple scattering events, showing good agreement between the theory and full-wave analysis under the first-order Born approximation. The result also verifies the experimental feasibility of the material phases described in Fig. 6, which can be implemented with optical or radio-frequency dielectric scatterers having geometric parameters accessible with conventional photolithography or direct laser writing.

## Discussion

The independent manipulation of short-range and long-range scattering provides immediate applications in photonics, phononics, solid-state physics and other fields related to scattering phenomena. For

example, because realizing hyperuniform patterns defined by structure factors is one of the necessary conditions for the complete bandgap by guaranteeing the unique existence of backscattering states, controlling short-range fluctuations while preserving suppressed long-range fluctuations can be applied to manipulate the reflection efficiency and defect-induced energy confinement in bandgap materials[61]. Super-scattering states of phases IV and V compared with crystals or Poisson materials also reveal effectively superdense material phases for waves with a given spatial range $\mathbf{\Omega}$ and particle number. Notably, the first-order Born approximation employed in this work restricts the range of materials that can be described by scattering networks. To extend the independent control of short-range and long-range order to strong scattering conditions, the concept of evolving scattering networks needs to be generalized to higher-order Born series to cover multiple and resonant scattering, and to full-vectorial wave equations for three-dimensional structures.

The demonstrated evolving scattering networks obtained from different preference functions are just a few illustrative examples. As widely studied in network science[52], the careful manipulation of evolution processes leads to notable changes in network topology and signal processing performances. Because higher preference can be considered the selective activation of particles according to their node degree, I envisage the use of widely used activation functions[62] to

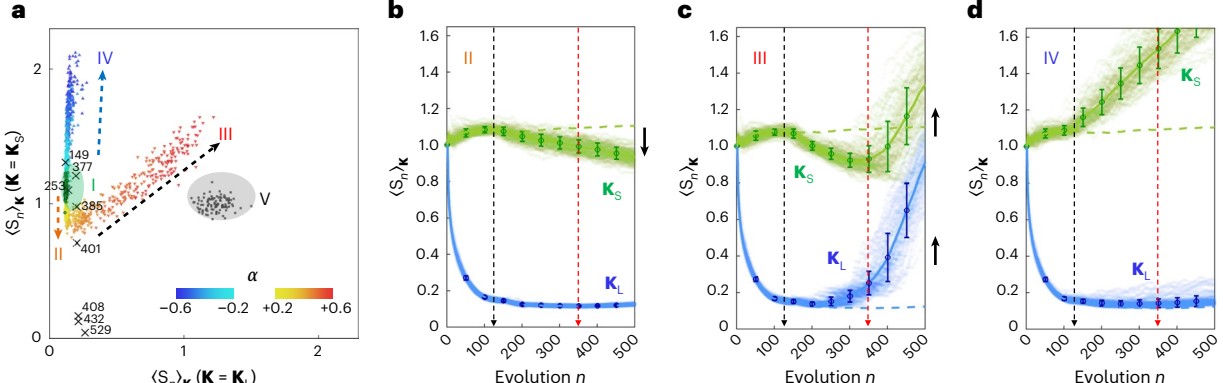

**Fig. 6 | Material phases of evolving scattering networks with preferential attachment. a**, Phase diagram defined by the scattering from $\mathbf{K}_L$ and $\mathbf{K}_S$. Each marker denotes a realization, and its color represents the form of preferential attachment characterized by $\alpha$ (from −0.6 to −0.2 and from +0.2 to +0.6 with 0.1 intervals). Phase I with green markers represent the SHU material with non-preferential attachment ($\alpha = 0$) in Fig. 3b. Phases II, III and IV have $\alpha = +0.2$, +0.6 and −0.6, respectively. Phase V with black markers represent the Poisson material in Fig. 3a. The cross markers denote crystal structures, and the numbers next to them indicate the numbers of the particles in each crystal. **b–d**, Evolutions of $\langle S_n \rangle_{\mathbf{K}_L}$ (blue points) and $\langle S_n \rangle_{\mathbf{K}_S}$ (green points) during the processes for the phases II (**b**), III (**c**) and IV (**d**). Black and red dashed arrows denote two branch points $n_{B1}$ and $n_{B2}$ in the material phase transition, respectively. Blue and green dashed lines denote the results of the SHU material with non-preferential attachment. Black arrows represent the relative change of scattering due to preferential and anti-preferential attachment. A random ensemble of 100 realizations each with 500 particles is investigated for all cases except the crystal. In **b–d**, circles and error bars represent the mean value and one standard deviation of each ensemble of 100 realizations.

the preference function $\Pi(w_p^{\mathbf{K}})$, such as softmax, rectified linear unit (ReLU), Gaussian error linear unit (GELU) or swish functions instead of tangent hyperbolic functions employed here. The target or initial material states do not have to be the SHU material inside the circular real space, which can be freely tunable by changing the designed reciprocal space $\mathbf{K}$ or real space $\mathbf{\Omega}$.

In terms of material science, network science and neural networks, there are many topics for future research on evolving scattering networks. In engineering disorder for waves[32], it is critical to retain the design space as broad as possible while preserving a specific wave quantity because the other wave quantities can be manipulated in the desired manner. Although I applied the deterministic rule to the cost function and achieved a random ensemble with different Monte Carlo realizations, the cost function can be directly employed to describe the probabilistic attachment of particles, which will allow further extension of design space for engineering disorder. In including particles to scattering networks, the finite space $\mathbf{\Omega}$ corresponds to the finite resource (or ability to generate links) the particle can utilize. Because the sequences of the evolution process gradually consume the resource, the evolution trend has to be changed in closed systems when increasing the particle number, which gives an insight to numerous similar situations in physics: fermionic systems with the Pauli exclusion and hysteresis responses with optical nonlinearity. In terms of realizing wave neural networks[17–23], the input (that is, incident waves) and output (that is, scattering waves) of the proposed scattering network are connected through the dynamical weight distribution (that is, $\mathbf{k}$-dependent interference), which is determined by evolving spatial ordering. Because all of the existing wave neural networks have treated static networks[17–23], and traditional evolutionary algorithms lack the network model for characterizing interacting neurons[34,35], the concept of evolving scattering networks inspires the realization of neuroevolution[12] in wave physics.

Evolving network models are not the exclusive path to disclosing the secrets of complex networks, as already demonstrated in scale-free networks obtained with the deterministic process[7] or static conditions[8]. Correspondingly, I can envisage various different static methodologies in achieving similar material states shown in this work, such as defining the cost function including short-range and long-range scattering while preserving the number of particles. However, as demonstrated in the

critical impact of evolving network model—unveiling the dynamics of time-varying networks—evolving scattering networks provide a multifaceted tool for engineering dynamical wave phenomena with a bridge to network science.

## Methods

### Assumptions in defining scattering networks

To focus on the effect of spatial ordering, I assume that each particle and the space between particles is composed of isotropic media, while the distribution of the particles $\mathbf{r}_j$ is inhomogeneous in general. When an incident wave with the wavevector $\mathbf{k}_I$ scatters off of a given material, I set the far-field measurement of the scattering wave having the wavevector $\mathbf{k}_S$ ($|\mathbf{k}_I| = |\mathbf{k}_S| = k_o$). As described in Supplementary Note 1, I also assume the weak scattering regime, allowing for the first-order Born approximation in the Lippmann–Schwinger equation.

### Classification of materials with impulse scattering responses

To classify materials in terms of their scattering responses[32], it is instructive to examine the impulse scattering response from $S_n(\mathbf{k}) = \delta(\mathbf{k} - \mathbf{k}')$, where $\delta(\mathbf{k})$ is the Dirac delta function. As an illustrative example, I show two-dimensional scattering problems in Fig. 1b–d (yellow points for $\pm \mathbf{k}'$). At a given wave frequency, a light cone (red solid circles in Fig. 1b–d) determined by the background material specifies the allowed states in reciprocal space. From the shift $\mathbf{k}_S = \mathbf{k}_I + \mathbf{k}$ and the reciprocity $S_n(\mathbf{k}) = S_n(-\mathbf{k})$, the scattering from $S_n(\mathbf{k}) = \delta(\mathbf{k} - \mathbf{k}')$ results in two $\mathbf{k}'$-shifted light cones (red dashed circles in Fig. 1b–d). Because scattering waves share the same light cone with the incident one, the intersections between the original and shifted light cones correspond to the allowed pairs of the incident and scattering wavevectors (red arrows in Fig. 1b–d) for the material of $S_n(\mathbf{k}) = \delta(\mathbf{k} - \mathbf{k}')$. The allowed $\mathbf{k}_S$ for $|\mathbf{k}'| < 2k_o$ is given by (Supplementary Note 2):

$$\mathbf{k}_S = \frac{1}{2}\left(\mathbf{I} + \left[\left(\frac{2k_o}{k'}\right)^2 - 1\right]^{1/2}\mathbf{R}_{\pm \pi/2}\right)\mathbf{k}', \tag{7}$$

where $\mathbf{I}$ and $\mathbf{R}_{\pm \pi/2}$ are the identity and $\pm \pi/2$ rotation operators, respectively. Equation (7) shows that impulse scattering responses are classified into three regimes: forward scattering ($|\mathbf{k}'| < \sqrt{2}k_o$ for $0 \le \theta \le \pi/2$

or $3\pi/2 \leq \theta \leq 2\pi$; Fig. 1b), backward scattering ($\sqrt{2}k_o \leq |\mathbf{k}'| < 2k_o$ for $\pi/2 \leq \theta \leq 3\pi/2$; Fig. 1c) and zero scattering ($|\mathbf{k}'| \geq 2k_o$; Fig. 1d) (Supplementary Note 2), where $\theta$ denotes the angle between incident and scattering waves.

## Criteria for evolving scattering networks

The first step to applying network science to wave phenomena is to define network parameters that can provide simplified and systematic interpretations of complex wave–matter interactions while maintaining physical validity. Although some guided[19,27,33] or diffractive[20] systems have been interpreted as network structures by defining the transmissive[19–21,33] or evanescent[21,27] links neglecting reflections, the network interpretation of scattering phenomena handling complex interferences from multiple particles is still challenging. For example, quantum graphs enable the network modeling of scattering through the graph edges defined by the metric graph between particles and the governing Hamiltonian, and the graph vertices for field boundary conditions[13–16]. Although the quantum graph model for scattering corresponds to the network interpretation of a rigorous scattering matrix, this rigorous modeling, at the same time, hinders the extraction of the kernel part of scattering networks, especially when participating elements are numerous, such as scattering from disordered materials.

I also note that all the previous network structures applied to wave phenomena[19–21,27,33], including quantum graphs[13–16], have employed static or generative models with time-independent network sizes, lacking the design principle for open systems that allow for material exchanges with the system environment. From the analysis of wave scattering using the Lippmann–Schwinger equation that allows for extracting kernel parts of scattering with the Born series[63], I develop an evolution model for wave–matter interactions as the analogy of evolving networks[1,51,52], which derives the suitable definition of wave networks for scattering phenomena.

## Evolution of the structure factor

From equation (1), I derive the evolution of the structure factor (Supplementary Note 3):

$$S_{n+1}(\mathbf{k}) = \frac{nS_n(\mathbf{k}) + 1}{n+1} + \frac{n}{n+1}\xi_n(\mathbf{k}, \mathbf{r}_{n+1}), \qquad (8)$$

where $\xi_n(\mathbf{k}, \mathbf{r}_{n+1})$ is the core function governing the evolution process with the following form:

$$\xi_n(\mathbf{k}, \mathbf{r}_{n+1}) = \frac{2}{n}\sum_{j=1}^{n}\cos\left[\mathbf{k}\cdot(\mathbf{r}_j - \mathbf{r}_{n+1})\right]. \qquad (9)$$

The material evolution then leads to the evolving change of the scattering intensity, as $I_{n+1}(\mathbf{k}; \mathbf{R}) = I_n(\mathbf{k}; \mathbf{R}) + I_1(\mathbf{k}; \mathbf{R})[1 + n\xi_n(\mathbf{k}, \mathbf{r}_{n+1})]$. Equations (8) and (9) present the underlying concept of evolving scattering networks. When $\xi_n(\mathbf{k}, \mathbf{r}_{n+1}) \approx 0$, equation (8) composes the recurrence relation $S_{n+1} = (nS_n + 1)/(n + 1)$, which gives $\lim_{n\to\infty} S_n = S_1 = 1$. Therefore, the alteration of $S_n(\mathbf{k})$ from the initial state originates from $\xi_n(\mathbf{k}, \mathbf{r}_{n+1})$ of which the cosine function represents the interference newly generated by the $(n + 1)$th particle. This result demonstrates the definition of the interference link $\cos[\mathbf{k}\cdot(\mathbf{r}_p - \mathbf{r}_q)]$ between the $p$th and $q$th nodes.

## Evolution process

To determine the position of a new particle, I introduce the $\mathbf{K}$-dependent cost function $\rho_n^{\mathbf{K}}(\mathbf{r})$ for the $n$th evolution, where the minimum of $\rho_n^{\mathbf{K}}(\mathbf{r})$ represents the best position for the $(n + 1)$th particle. As an analogy of the evolution process in network science[1,2,52], $\rho_n^{\mathbf{K}}(\mathbf{r})$ should reflect the network connectivity after the evolution and also include the rule for the preference, for example, the preferential attachment to the particles having higher node degrees. First, when a new particle is deposited at $\mathbf{r}$, the link weight between the $p$th particle and a new

particle is $(1/V_{\mathbf{K}})\int_{\mathbf{K}}\cos\mathbf{k}\cdot(\mathbf{r}_p - \mathbf{r})\,d\mathbf{k}$ according to equation (2), quantifying the network connectivity after the $n$th evolution. Second, because it is natural to determine the preference for the $p$th particle with its node degree[1,2,52], I define the preference function as $\Pi(w_p^{\mathbf{K}})$, which characterizes the preferential attachment to the $p$th node. The cost function $\rho_n^{\mathbf{K}}(\mathbf{r})$ is then defined by using two terms listed above, leading to equation (4).

## Generalization to inhomogeneous materials

In the evolving wave-network modeling of scattering phenomena, I apply the identical point particle assumption. This assumption provides an excellent insight due to its simplicity while preserving a good level of modeling in the regime of the first-order Born approximation with homogeneous constituents. For a more rigorous description of evolving scattering networks and a better accuracy of the modeling, I also extend the evolving network concept to inhomogeneous materials in Supplementary Note 4, generalizing the link weight and node degree of evolving scattering networks, the cost function for the evolution process, and their relations to the structure factor and inhomogeneous wave scattering. The result of Supplementary Note 4 demonstrates that the concept of evolving scattering networks is also valid for inhomogeneous materials. Notably, the inhomogeneity and finite sizes of particles are reflected in network parameters by the cross-correlation and autocorrelation of the potential landscapes, respectively, as shown in Supplementary Table 1.

## Preconditions and parameters for evolution processes

Because of the positive initial state $S_1 = 1$ from equation (1), I focus on the evolving suppression of wave scattering using equation (4), targeting the minimization of the cost function: finding $\mathbf{r}_{n+1} = \mathbf{r}_{min}$ for $\min[\rho_n^{\mathbf{K}}(\mathbf{r})] = \rho_n^{\mathbf{K}}(\mathbf{r}_{min})$ when $\Pi(w_p^{\mathbf{K}}) \geq 0$. In analyzing non-preferential and preferential attachment for evolving scattering networks (Figs. 2–6), I set the entire spatial domain $\mathbf{\Omega}^{tot}$ to be the circle of radius $R_{max}$. For an $n$-particle scattering network, the average unit area of each particle becomes $\pi R_{max}^2/n$, which determines the characteristic radius $r_c = R_{max}/n^{1/2}$ and the characteristic distance $d_c = 2r_c$. I then set the reciprocal space $\mathbf{K}^{tot}$ to be the circle of radius $k_c = 2\pi/d_c$, where $k_c$ is the upper limit of the spatial frequency of interest in calculating the structure factor. In the optimization process, I divide the real space $\mathbf{\Omega}^{tot}$ into the region of interest $\mathbf{\Omega}$ and its complementary space $\mathbf{\Omega}^c$, while the reciprocal space $\mathbf{K}^{tot}$ is also divided into the region of interest $\mathbf{K}$ and its complementary space $\mathbf{K}^c$. A newly included particle is located inside the space $\mathbf{\Omega}$, where the specific position is determined to minimize the scattering inside the reciprocal space $\mathbf{K}$.

## Specific values of parameters in classifying materials

In Fig. 3, I investigate the uncorrelated Poisson materials and evolving SHU materials having $n = 500$ particles for a single realization of $R_{max} = 1$ and $d_c = 0.0894$. The Poisson materials are achieved by randomly selecting the positions of $n$ particles from the $10^4$ candidate positions in the real space obtained by the Monte Carlo method. In constructing evolving SHU materials, I focus on the SHU condition satisfying $S_n(\mathbf{k}) \approx 0$ for $|\mathbf{k}| < 0.50k_c$, where $k_c = 70.248$. To reflect numerically insuppressible scattering of near-infinite wavelengths, I set $k_{min} = 0.05k_c$, resulting in $\mathbf{K}_L = \{\mathbf{k} \mid 0.05k_c \leq |\mathbf{k}| \leq 0.50k_c\}$ and $\mathbf{K}_S = \{\mathbf{k} \mid 0.50k_c < |\mathbf{k}| \leq k_c\}$. The 100 realizations of each ensemble are obtained by applying the Poisson process and evolution process to different realizations of the real-space Monte Carlo discretization by controlling the seed for pseudorandom number generators. I employ different seeds also for the reciprocal-space Monte Carlo discretization in constructing evolving SHU materials, examining 2,475 reciprocal states of $\mathbf{K}_L$ in calculating equation (4). The crystal has the $x$ and $y$ periodicity of $\Lambda = 1/7$ for $R_{max} = 1$, possessing 149 particles inside $\mathbf{\Omega}^{tot}$. The periodicity is determined to obtain the first Bragg peak $|\mathbf{k}_{Bragg}| = 2\pi/\Lambda = 0.626k_c$ inside $\mathbf{K}_S$, to achieve similar scattering responses with those of evolving materials, as $S_n(\mathbf{k} \in \mathbf{K}_L) \approx 0$ and

$S_n(\mathbf{k} \in \mathbf{K}_s) \gg 0$. The properties of other crystals with different periodicities are discussed in Supplementary Note 9.

## Comparison with traditional approaches

The traditional strategy to achieve hyperuniform point patterns is the collective coordinate method[37,57,59] or its extension to molecular dynamics[38,60], which minimizes the potential energy defined by particle positions and the wavevector $\mathbf{k}$. Compared with the evolving scattering network model, the collective coordinate method does not explicitly consider an underlying network structure for wave physics. Furthermore, because the potential energy in the collective coordinate method is defined for the system of a fixed number of particles, the method does not allow varying particle numbers in its current implementation, similar to generative models in network science[52], which update the network structure according to the predefined rule for degree distribution. Considering the impacts of evolving network models compared with generative models[52], such as characterizing time-varying network topologies and revealing the origin of network properties, I explore the SHU evolution process to achieve the 'screening' of existing materials in the main text.

## Specific values of parameters in evolving material screening

In Fig. 4, I investigate the SHU mixing, core and cladding configurations for $n = 500$ particles in a single realization, which is composed of 300 particles obtained from the SHU process and 200 particles obtained from the Poisson process. In the SHU mixing (Fig. 4a,d), the entire real space $\mathbf{\Omega}^{tot}$ is shared with the SHU and Poisson processes. In the core and cladding configurations, I define the real-space subsets of $\mathbf{\Omega}^{tot}$: $\mathbf{\Omega}$ for the SHU evolution and $\mathbf{\Omega}^c$ for the Poisson evolution. Each subset is defined to maintain the statistical density of particles over the entire space, by dividing $\mathbf{\Omega}$ (red shaded area) and $\mathbf{\Omega}^c$ (blue shaded area) with the circle boundaries of the radii 0.7746 and 0.6325 for core and cladding configurations, respectively.

## Data availability

The datasets for particle distributions, structure factors, and network parameters supporting the findings of this study are available at https://doi.org/10.5281/zenodo.7471426 in Zenodo[64]. Source data are provided with this paper.

## Code availability

Core codes developed in this work have been deposited at https://doi.org/10.24435/materialscloud:ym-kr in the Materials Cloud Archive[65]. The codes were developed with MATLAB R2021a. Supplementary codes for Figs. 2–6 are available with this paper.

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

## Acknowledgements

I acknowledge financial support from the National Research Foundation of Korea (NRF) through the Basic Research Laboratory (No. 2021R1A4A3032027), Young Researcher Program (No. 2021R1C1C1005031) and Global Frontier Program (No. 2014M3A6B3063708), all funded by the Korean government.

## Author contributions

S.Y. conceived the idea, discussed the results and contributed to the final manuscript.

## Competing interests

The author declares no competing interests.

## Additional information

**Correspondence and requests for materials** should be addressed to Sunkyu Yu.

