## [Peer Review File · Nature Computational Science]

Peer Review Information

Journal: Nature Computational Science

Manuscript Title: Evolving scattering networks for engineering disorder

Corresponding author name(s): Sunkyu Yu

Editorial Notes:

Reviewer Comments & Decisions:

Decision Letter, initial version:

Dear Professor Yu,

Your manuscript "Evolving wave networks for engineering disorder" has now been seen by 3 referees, whose comments are appended below. You will see that while they find your work of interest, they have raised points that need to be addressed before we can make a decision on publication.

The referees' reports seem to be quite clear. Naturally, we will need you to address all of the points raised.

While we ask you to address all of the points raised, the following points need to be substantially worked on:

- Compare (qualitatively *and* quantitatively if possible) your proposed method against existing methods for hyperuniform materials, scattering methods, quantum graphs, etc., as requested by Referees #2 and #3.
- Better clarify the novelty of your proposed method when compared to other methods.
- Better clarify and define the terms used in the paper, such as "wave", "wave network", and "evolving network", to avoid confusion.
- Add benchmark results beyond the Born approximation region to address point 2 from Referee #1.

Please use the following link to submit your revised manuscript and a point-by-point response to the referees' comments (which should be in a separate document to any cover letter):

[REDACTED]

** This url links to your confidential homepage and associated information about manuscripts you may have submitted or be reviewing for us. If you wish to forward this e-mail to co-authors, please delete this link to your homepage first. **

To aid in the review process, we would appreciate it if you could also provide a copy of your manuscript files that indicates your revisions by making use of Track Changes or similar mark-up tools. Please also ensure that all correspondence is marked with your Nature Computational Science reference number in the subject line.

In addition, please make sure to upload a Word Document or LaTeX version of your text, to assist us in the editorial stage.

To improve transparency in authorship, we request that all authors identified as 'corresponding author' on published papers create and link their Open Researcher and Contributor Identifier (ORCID) with their account on the Manuscript Tracking System (MTS), prior to acceptance. ORCID helps the scientific community achieve unambiguous attribution of all scholarly contributions. You can create and link your ORCID from the home page of the MTS by clicking on 'Modify my Springer Nature account'. For more information please visit www.springernature.com/orcid.

We hope to receive your revised paper within three weeks. If you cannot send it within this time, please let us know.

Best,
Fernando

--

Fernando Chirigati, PhD
Chief Editor, Nature Computational Science
Nature Portfolio

Reviewers comments:

Reviewer #1 (Remarks to the Author):

Dear Editor,

Please find enclosed my comments on the manuscript "Evolving wave networks for engineering disorder" by Yu, submitted for publication in Nature Computational Science.

The paper presents a comprehensive exploration of network-based classification of microstructured material and preferential attachment during material-design evolution. There are a number of interesting results and ideas put forward including a novel way to generate materials with certain

scattering properties, including stealthy hyperuniform disordered structures, open-system material design and “super-dense” material phases for scattering.

The study presented is very comprehensive with many numerical experiments performed and discussed in great detail; the manuscript is also generally well-written (although the presentation is fragmented at times) and, in my view, is well-suited for publication in Nature Computational Science.

There are a couple of important issues that need addressing:

1. Throughout the manuscript (including the title) is mentioned the syntagm “wave networks”. At face value (as mentioned above and noted in the manuscript), what the author does is to develop a “network-based” exploration/design/classification of disordered materials. While I fully agree that the network concepts are essential to the whole formalism, I find it misleading here as it can be easily confused with physical networks which are of relevance of the many applications mentioned in the manuscript (e.g. photonic band gap materials). Not to mention that for materials displaying physical networks (e.g. 2D TE polarization or 3D photonic band gap materials), the current structure factor-based analysis is not appropriate. The author will need to address this aspect in the manuscript and make sure there is no confusion possible.
2. In the same note, the other word in the syntagm, “wave” is also a source of possible confusion. Throughout the manuscript, the author works in the Born approximation, a rather restrictive regime of scattering. While there are scattering physical phenomena where the Born approximation is well justified, many of the applications which seem to be the focus of the discussion in the manuscript (again band gap materials, be they electronic, acoustic/photonic or photonic) strongly rely upon multiple scattering and resonance scattering where the formalism developed here fails. The author does present a comparison between results under the Born approximation and “full-wave” simulations, but the agreement is unsurprising since the parameters were chosen to set the system in the Born-approximation regime. To make my argument clearer, current study cannot predict if a conventional band gap opens in the structures designed, at what frequency it may open or what its size would be. Indeed, the gap eventually occurs due to backscattering states, but their nature is much more complex than presented here and resonance scattering, and multiple scattering play an essential role impossible to be captured in the simple model of scattering presented here. Again, the author would need to clarify this aspect and I would suggest replacing the word “wave” with “scattering” or similar.

Reviewer #2 (Remarks to the Author):

The manuscript entitled “Evolving wave networks for engineering disorder” describes theoretical evolving wave networks, which address networks whose connections and number of nodes change in time, and which the author uses to describe complex and open physical systems. Moreover, a specific design property is looked for, i.e. a special correlation, the so called hyper uniformity.

The manuscript is well-written, detailed, and addresses the growing field of wave network, as the author calls it. Network theory is transforming the way we model complex phenomena, and its application to physical systems, from complex materials to nonlinear interaction is growing.

The key novelty here is the addition of “evolution”, i.e. time dependence. I had therefore high

expectation of seeing some dynamical properties, while instead the manuscript describes algorithms to developed large networks adding a particle at a time. This is not new, it is similar to molecular dynamics approaches, and to the way hyper uniform materials have been designed so far (eg <https://doi.org/10.1073/pnas.1705130114>, or <https://doi.org/10.1364/OPTICA.3.000763>). I also find the manuscript full of text books results, eg where the structure factor is derived in Results, or figure 3, which are not new and should go to supplementary material. The network approach used in the introduction is soon lost in the manuscript, and I am not sure what new science I am learning from it, and further more which computational science.

Therefore I cannot recommend it for publication in Nature Computer Science and instead I would recommend it for a more specialised journal.

More in details

1. The author never mentions quantum graphs, which are very close to wave networks. I believe they should be compared and introduced.
2. Evolving networks is the key concept, but it is never defined.
3. what is the difference of this network method and just a traditional scattering method? Which network property is used here, eg in fig 3?
4. can the proposed method outperform current ones?

Reviewer #3 (Remarks to the Author):

In this paper, the author developed a novel wave-network scheme for the design and optimization of photonic network materials, which was demonstrated in stealthy hyperuniform material systems. In particular, the author formulated the material optimization problem, in which the materials are composed of a varying number of isotropic scatters, into an evolving wave network in the context of scattering. One key novelty of the work is the proposed mapping between network science and wave physics, through the definition of nodes, weighted links, degree distributions, and evolution processes based on scattering theory. The utility of the framework was demonstrated by designing and achieving extraordinary material states such as effectively denser or sparser particle distributions for short-range order while preserving long-range order of crystals or Poisson processes. This approach provides the network-based interpretation of wave phenomena, extending the candidate platforms for wave neural networks.

The paper is overall very well written and of great topical interest. The mapping between network and wave physics, as well as the concept using evolving network for photonic material design are very novel and inspiring. A key advantage compared to previously work on photonic material optimization and hyperuniform material design is that the current work allows varying particle numbers, enabling multi-scale open material system design. I'm happy to recommend its publication once the author addresses the following comments and suggestions for minor revisions:

Although viewing material scattering as a wave network is very inspiring, the key quantity of interest that characterizes the scattering behavior is the statistic structure factor, which was the objective function employed in the optimization. In this regard, the current approach, especially on the

fundamental mathematical level, bears similarity with the collective coordinate approach developed by Torquato and co-workers, which were discussed in many of the references cited in the paper. The key differences between the two include: (i) the collective coordinate approach does not explicitly consider an underlying “network” for the wave physics (ii) it does not allow varying particle numbers in its current implementation. It might be useful for the author to elaborate these points a little in the paper.

Some minor points:

P3, L65, “...suggested network modeling...” should be “... suggested network model...”?

In Fig 1, the network modeling for scattering process was not very apparent, as the links were not shown?

Author Rebuttal to Initial comments

Reply to Editorial Requests

*- Compare (qualitatively *and* quantitatively if possible) your proposed method against existing methods for hyperuniform materials, scattering methods, quantum graphs, etc., as requested by Referees #2 and #3.*

The novelty of my work is based on “network model” and “evolution process” applied to *wave physics*. Both properties are essential for realizing evolving scattering networks. I included the following discussion and Table 1 to compare my proposal with several traditional methods.

(i) Collective coordinate method

The collective coordinate method developed by Torquato and co-workers [refs 37, 60] allows for designing hyperuniform materials by minimizing the potential energy obtained with the simplified modelling of scattering. **The key novelty of my work against this method is (i-1) the realization of the “network model” characterized by scattering and (i-2) the application of the evolution with “varying” particle numbers.** The potential energy in the collective coordinate method has been defined for a *fixed* number of particles.

(ii) Molecular dynamics

Molecular dynamics for designing hyperuniformity is the extension of the collective coordinate method, which is based on the potential energy minimization achieved by *molecular dynamics* with the equation of motion. **Therefore, the key novelty of my work from this method is the same as those in (i) because the potential energy is again defined for a fixed number of particles.**

(iii) Quantum graphs

The key novelty of my work against quantum graphs is (i) the extraction of the network kernel and (ii) the evolution process.

First, quantum graphs correspond to the network-based modelling of *rigorous* scattering matrix methods based on *full* interactions between wave nodes, **which hinders the extraction of the kernel part of wave networks** when participating elements are numerous. **The solution for this challenging issue is to utilize the traditional approach in scattering problems: using the “approximated” Born series.** Although the Born series and the differential wave equation are mathematically identical, **the Born series enables the systematic approximation of scattering, allowing for extracting the network kernel using the first-order Born approximation.** In the revised manuscript, I clarified such a novelty of my work against quantum graph theory.

(Lines 48-53) Notably, quantum graphs provide an analytical tool to fully describe the interactions between wave nodes¹³⁻¹⁶. However, extracting the kernel of a wave network requires platform-specific simplification, as shown in the reflectionless and single-channel assumptions for the network modelling of guided^{19,27,33} and diffractive²⁰ systems. In this context, the efficient network modelling of scattering phenomena having complex interferences from multiple particles is still an open question.

(Lines 147-153) For example, quantum graphs enable the network modelling of scattering through the graph edges defined by the metric graph between particles and the governing Hamiltonian, and the graph vertices for field boundary conditions¹³⁻¹⁶. Although the quantum graph model for scattering corresponds to the network interpretation of a rigorous scattering matrix, this rigorous modelling, at the same time, hinders the extraction of the kernel part of wave networks, especially when participating elements are numerous, such as scattering from disordered materials.

(Lines 157-159) From the analysis of wave scattering using the Lippmann-Schwinger equation that allows for extracting kernel parts of scattering with the Born series⁵¹, I develop an evolution model for ...

Second, another novelty of my work is the “evolution”, which has not been studied in quantum graphs. **There exist only a few examples of handling time-varying quantum graphs, which are not about the networks with varying network sizes but about time-varying unitary evolution [Наносистемы: физика, химия, математика 6, 173 (2015)].** I clarified this point.

(Lines 154-156) I also note that all the previous network structures applied to wave phenomena^{19-21,27,33}, including quantum graphs¹³⁻¹⁶, have employed static or generative models with time-independent ...

(iv) Wave neural networks

Wave neural networks have been implemented with different platforms: the cascaded unitary and diagonal operators realized with phase delay lines and beam splitters [ref. 19], or multiple diffractive layers [ref. 20]. **The concept of wave neural networks has not been applied to both the evolution process and the design of hyperuniformity.**

Table 1. The comparison of traditional design methods and evolving scattering networks. The properties of interest denote the necessary conditions for applying evolving networks to wave physics, while preserving the advantages of network science: simplified and systematic understanding of complex phenomena. Here, “No” does not mean “impossible”. It may be close to the answer to the question about the explicit implementation of the property. For example, although quantum graphs may be implemented with the evolution process, such an approach is still absent.

	Network modelling	Evolution	Preference	Simplified modelling	Hyperuniformity Design
Collective coordinate	No	No	No	Yes	Yes
Molecular dynamics	No	No	No	Yes	Yes
Quantum graph	Yes	No	No	No	No
Wave neural networks	Yes	No	No	Yes	No
Evolving scattering network	Yes	Yes	Yes	Yes	Yes

Based on the listed comparisons, I carefully revised the manuscript, as follows:

(Lines 338-345) The traditional strategy to achieve hyperuniform point patterns is the collective coordinate method^{37,58,60} or its extension to molecular dynamics^{38,61}, which minimizes the potential energy defined by particle positions and the wavevector \mathbf{k} . Compared with the evolving scattering network model, the collective coordinate method does not explicitly consider an underlying network structure for wave physics. Furthermore, because the

potential energy in the collective coordinate method is defined for the system of a *fixed* number of particles, the method does not allow varying particle numbers in its current implementation, as similar to generative models in network science⁵³, ...

(Lines 392-395) ... the preferential attachment is a dynamical process, in sharp contrast to the static or generative methodologies with preassigned rules, such as the collective coordinate method^{37,58,60} or its extension to molecular dynamics^{38,61}.

- *Better clarify the novelty of your proposed method when compared to other methods: what does your method allow, more broadly speaking, that others do not?*

First, compared with a traditional scattering method (e.g. collective coordinate approach) or evolutionary algorithms, the critical difference is the network concept based on wave interferences. Although traditional approaches have been studied for designing wave scattering, these methods include neither the network viewpoint nor open-system natures—dynamically changing the system size. Therefore, it is difficult to examine the impact of an individual network element in the traditional methods, lacking the bridge between scattering phenomena and wave neural networks that require individual elements, i.e. “neurons”. I revised the manuscript to clarify the above discussion.

(Lines 534-537) Because all of the existing wave neural networks have treated static networks¹⁷⁻²³, and traditional evolutionary algorithms lack the network model for characterizing interacting neurons^{34,35}, the concept of evolving scattering networks inspires the realization of neuroevolution¹² in wave physics.

Second, the advantage of the proposed method is also clear when handling *dynamically-varying scattering systems*, as shown in Fig. 4—the first demonstration of material screening with SHU. For open-system problems, such as the alteration of a given system with the additional inclusion of elements (e.g., Fig. 4), the evolution process is a more superior strategy, as shown in the SHU conservation with evolving network models in Fig. 4. Although the final states of a material (or a network) obtained from those two approaches may not always be different, **the evolution process can also reveal unexplored states, as shown in the discovery of super-dense material phases in shortrange order in Figs 5 and 6 by introducing the “preference” for material design for the first time.**

- *Better clarify and define the terms used in the paper, such as “wave”, “wave network”, and “evolving network”, to avoid confusion.*

I replaced the term “wave networks” with “scattering networks” in the revised manuscript (including the title) to clarify the impact of my work. I also included the definitions of “evolving network models” and “evolving scattering networks”:

(Lines 23-24) Evolving network models—the models that characterize the mechanisms and natures of timevarying networks—have stimulated significant advances in network science and related disciplines.

(Lines 65-67) Here, I propose the concept of evolving scattering networks—the open-system wave network models with a dynamically changing number of particles inside a system—which provides a novel tool for multiscale material design with target scattering responses.

- Add benchmark results beyond the Born approximation region to address point 2 from Referee #1.

As Referee #1 commented, my work is suitable to engineer weak scattering rather than reproducing multiple-scattering-based results, such as bandgaps. The engineering of hyperuniform patterns using evolving scattering networks is close to the necessary but not sufficient condition for bandgap physics. The validity of the first-order Born approximation compared to the full-wave analysis has been thoroughly studied in previous literature ([ref. 51; J. Comput. Chem. 23, 1297 (2002); Prog. Electromagn. Res. C 107, 219 (2010)]), and **the range of validity for my work was shown in Supplementary Notes S14** (“This condition satisfies the first-order Born approximation roughly for the freespace optical wavelength of $\lambda_o > 0.56\lambda_{sim} = 847 \text{ nm}$ (or, $k_o = 2\pi/\lambda_o < 1.77k_{sim} = 0.95k_c$).”).

I agree that **further generalization of evolving networks to higher-order Born series and fullvectorial wave equations is essential.** I revised the manuscript following Referee #1’s comment.

(Lines 498-499) To extend such intriguing features to strong scattering conditions beyond the first-order Born approximation, the concept of evolving wave networks needs to be generalized to higher-order Born series to cover multiple and resonant scattering and to full-vectorial wave equations for 3D structures.

(Lines 504-507) Because realizing hyperuniform patterns defined by structure factors is one of the necessary conditions for the complete bandgap by guaranteeing the unique existence ...

Reply to Reviewer 1’s report

Dear Editor,

Please find enclosed my comments on the manuscript “Evolving wave networks for engineering disorder” by Yu, submitted for publication in Nature Computational Science.

The paper presents a comprehensive exploration of network-based classification of microstructured material and preferential attachment during material-design evolution. There are a number of interesting results and ideas put forward including a novel way to generate materials with certain scattering properties, including stealthy hyperuniform disordered structures, open-system material design and “super-dense” material phases for scattering.

The study presented is very comprehensive with many numerical experiments performed and discussed in great detail; the manuscript is also generally well-written (although the presentation is fragmented at times) and, in my view, is well-suited for publication in Nature Computational Science.

I sincerely appreciate the reviewer’s professional and positive comments on my manuscript. Following the reviewer’s suggestions, I carefully revised my manuscript and now believe that my manuscript was greatly improved.

There are a couple of important issues that need addressing:

1. Throughout the manuscript (including the title) is mentioned the syntagm “wave networks”. At face value (as mentioned above and noted in the manuscript), what the author does is to develop a “network-based” exploration/design/classification of disordered materials. While I fully agree that the network concepts are essential to the whole formalism, I find it misleading here as it can be easily confused with physical networks which are of relevance of the many applications mentioned in the manuscript (e.g. photonic band gap materials). Not to mention that for materials displaying physical networks (e.g. 2D TE polarization or 3D photonic band gap materials), the current structure factor-based analysis is not appropriate. The author will need to address this aspect in the manuscript and make sure there is no confusion possible.

Thank you very much for your comment. In the revised manuscript, I clarified that the concept of “wave networks” is distinct from “physical networks”, which correspond to the networks defined by the composition of material or structural design.

(Lines 40-41) In addition to understanding physics in material or structural networks, realizing networks defined by wave-matter interactions has received considerable attention.

The limitation of the current structure-factor-based analysis was also clarified in the Discussion section (shown in the reply to Point 2). Also, as shown in the reply to Point 2, I replaced the term “wave networks” with “scattering networks” to accurately describe my results.

2. In the same note, the other word in the syntagm, “wave” is also a source of possible confusion. Throughout the manuscript, the author works in the Born approximation, a rather restrictive regime of scattering. While there are scattering physical phenomena where the Born approximation is well justified, many of the applications which seem to be the focus of the discussion in the manuscript (again band gap materials, be they electronic, acoustic/photonic or photonic) strongly rely upon multiple scattering and resonance scattering where the formalism developed here fails. The author does present a comparison between results under the Born approximation and “full-wave” simulations, but the agreement is unsurprising since the parameters were chosen to set the system in the Born-approximation regime. To make my argument clearer, current study cannot predict if a conventional band gap opens in the structures designed, at what frequency it may open or what its size would be.

Indeed, the gap eventually occurs due to backscattering states, but their nature is much more complex than presented here and resonance scattering, and multiple scattering play an essential role impossible to be captured in the simple model of scattering presented here. Again, the author would need to clarify this aspect and I would suggest replacing the word "wave" with "scattering" or similar.

I completely agree with the reviewer’s opinion on the terminology “wave networks”. As the reviewer stated, my work is based on the Born approximation for scattering problems, and therefore, it is suitable to engineer weak scattering rather than reproducing multiple-scattering-based results, such as bandgaps. **In this context, the engineering of hyperuniform patterns using evolving scattering networks is close to the necessary but not sufficient condition for bandgap physics.** To handle bandgap phenomena, further generalization of the concept of evolving networks to higher-order Born series and full-vectorial wave equations is essential, which will be a future research topic.

Following the reviewer’s comments, I replaced the term “wave networks” with “scattering networks” in the revised manuscript (including the title) to clarify the impact of my work. I also carefully revised the manuscript following the reviewer’s suggestion.

(Lines 498-499) To extend such intriguing features to strong scattering conditions beyond the first-order Born approximation, the concept of evolving wave networks needs to be generalized to higher-order Born series to cover multiple and resonant scattering and to full-vectorial wave equations for 3D structures.

(Lines 504-507) Because realizing hyperuniform patterns defined by structure factors is one of the necessary conditions for the complete bandgap by guaranteeing the unique existence ...

Reply to Reviewer 2's report

The manuscript entitled "Evolving wave networks for engineering disorder" describes theoretical evolving wave networks, which address networks whose connections and number of nodes change in time, and which the author uses to describe complex and open physical systems. Moreover, a specific design properties is looked for, i.e. a special correlation, the so called hyper uniformity.

The manuscript is well-written, detailed, and address the growing field of wave network, as the author calls it. Network theory is transforming the way we model complex phenomena, and its application to physical systems, from complex materials to nonlinear interaction is growing.

I sincerely appreciate the reviewer's positive comments on the topic and presentation of my manuscript.

In the revision, I have focused on clarifying the novelty of my work following the reviewer's comments.

The key novelty here is the addition of "evolution", i.e. time dependence. I had therefore high expectation of seeing some dynamical properties, while instead the manuscript describes algorithms to developed large networks adding a particle at a time. This is not new, it is similar to molecular dynamics approaches, and to the way hyper uniform materials have been designed so far (eg <https://doi.org/10.1073/pnas.1705130114>, or <https://doi.org/10.1364/OPTICA.3.000763>).

Thank you very much for your comment, which is certainly helpful in clarifying the novelty of my work.

The novelty of my work is based on the following two features: "network model" and "evolution process" for wave physics. Both properties are essential for realizing evolving networks of scattering phenomena. In this context, I included the discussion and Table 1 in this reply letter to compare my proposal with traditional design methods or other related topics.

(i) Collective coordinate method

The suggested Optica paper [Optica 3, 763-767 (2016): ref. 58] employed the collective coordinate method developed by Torquato and co-workers [Phys. Rev. E 70, 046122 (2004); J. Appl. Phys. 104, 033504 (2008): refs 37, 60]. This method leads to the design of hyperuniform materials by minimizing the potential energy that is obtained with the simplified modelling of scattering. **The key novelty of my work against this method is (i-1) the realization of the network modelling defined by scattering properties and (i-2) the application of the evolution process with varying particle numbers.** Notably, the potential energy in the collective coordinate method has been defined for a *fixed* number of particles.

(ii) Molecular dynamics

Molecular dynamics employed in the suggested PNAS paper [Proc. Natl. Acad. Sci. 114, 95709574 (2017): ref. 61] is the extension of the collective coordinate method for hyperuniformity, which was also proposed by Torquato and co-workers [Phys. Rev. X 5, 021020 (2015): ref. 38]. This method is based on the potential energy minimization achieved by *molecular dynamics* with the equation of motion. **Therefore, the key novelty of my work from this method is the same as those in (i) because the potential energy is again defined for a fixed number of particles.**

(iii) Quantum graphs

: The novelty against quantum graphs is discussed in the reply to Point 1 in below.

(iv) Wave neural networks

Wave neural networks have attracted significant attention for machine learning or quantum computing. The networks have been implemented with different platforms: the cascaded unitary and diagonal operators realized with phase delay lines and beam splitters [Nat. Photon. 11, 441 (2017): ref. 19], or multiple diffractive layers [Science 361, 1004-1008 (2018); ref. 20]. Wave phenomena in these platforms are usually simplified for network modelling, for example, by neglecting reflections to achieve the unitary condition. **The concept of wave neural networks has not been applied to both the evolution process and the design of hyperuniformity.**

Table 1. The comparison of traditional design methods and evolving scattering networks. The properties of interest denote the necessary conditions for applying evolving networks to wave physics, while preserving the advantages of network science: simplified and systematic understanding of complex phenomena. Here, “No” does not mean “impossible”. It may be close to the answer to the question about the explicit implementation of the property. For example, although quantum graphs may be implemented with the evolution process, such an approach is still absent.

	Network modelling	Evolution	Preference	Simplified modelling	Hyperuniformity Design
Collective coordinate	No	No	No	Yes	Yes

Molecular dynamics	No	No	No	Yes	Yes
Quantum graph	Yes	No	No	No	No
Wave neural networks	Yes	No	No	Yes	No
Evolving scattering network	Yes	Yes	Yes	Yes	Yes

Based on the listed comparisons, I carefully revised the manuscript, as follows:

(Lines 338-345) The traditional strategy to achieve hyperuniform point patterns is the collective coordinate method^{37,58,60} or its extension to molecular dynamics^{38,61}, which minimizes the potential energy defined by particle positions and the wavevector \mathbf{k} . Compared with the evolving scattering network model, the collective coordinate method does not explicitly consider an underlying network structure for wave physics. Furthermore, because the potential energy in the collective coordinate method is defined for the system of a *fixed* number of particles, the method does not allow varying particle numbers in its current implementation, as similar to generative models in network science⁵³, ...

(Lines 392-395) ... the preferential attachment is a dynamical process, in sharp contrast to the static or generative methodologies with preassigned rules, such as the collective coordinate method^{37,58,60} or its extension to molecular dynamics^{38,61}.

I also cordially emphasize that the dynamical features of the proposed scattering networks were studied in the original manuscript. The key dynamics of evolving networks are the time-varying network structures and properties obtained with the evolution process, e.g., the inclusion or annihilation of network nodes. **After bridging wave parameters and network parameters in the section of “Evolving scattering networks”, the results in Fig. 4g,h and Fig. 6b-d apparently show the dynamical evolution of the network parameters—node degrees, as shown in Eq. (6)—in the proposed evolving scattering network. I clarified this point in the revised manuscript.**

(Lines 364-366) Because $\langle S_n \rangle_K$ directly reflects the average node degrees in K , Fig. 4g,h represents the dynamical evolutions of scattering network structures in terms of long-range and short-range order, which show the ...

I also find the manuscript full of textbook results, eg where the structure factor is derived in Results, or figure 3, which are not new and should go to supplementary material. The network approach used in the introduction is soon lost in the manuscript, and I am not sure what new science I am learning from it, and further more which computational science.

I'd like to emphasize that the results of Fig. 3 and the analysis of the structure factor with network parameters are critical for understanding the central claim of this manuscript. First, Fig. 3d-f and Fig. 3g-i denote the network parameters—node degree distributions—which are the conventional quantities for characterizing network structures [Barabási, A.-L. Network science (Cambridge university press, 2016)]. The traditional material classification and its network-based interpretation are apparently not the same things, as similar to the novelty and insight of quantum graphs that interpret known quantum phenomena. Therefore, I cordially claim that these node degree distributions are *not* the textbook results, as shown in the original manuscript (“(Lines 296-299) Although Fig. 3a-c shows a well-known structure factor of each material phase, Fig. 3d-i demonstrates that node degree distributions of evolving scattering networks operate as a useful tool for characterizing microstructures at the length scale of interest, bridging network analysis and material statistics.”).

Furthermore, Fig. 3j-l—the visualization of node degrees—are essential to understand the results of Fig. 4a-f and Fig. 5a,d,g, because Fig. 3j-l shows which elements are important in the network modelling, and this property is critical in understanding the effect of preferential attachments in the evolution process. Therefore, I decided to maintain the current form of Fig. 3.

Therefore I cannot recommend it for publication in Nature Computer Science and instead I would recommend it for a more specialised journal.

Following the reviewer's helpful suggestions and comments especially for the comparison with traditional methods (e.g., quantum graph and molecular dynamics), I carefully revised my manuscript and now firmly believe that my manuscript was greatly improved. I cordially expect your kind reconsideration of my manuscript.

More in details

1. The author never mentions quantum graphs, which are very close to wave networks. I believe they should be compared and introduced.

I sincerely appreciate the reviewer's insightful comment revealing the connection between my work and quantum graphs. I agree that the proposed scattering network is closely related to quantum graphs, which stems from the viewpoint on decomposing physical phenomena into interacting network nodes (or graph vertices) through network links (or graph edges). **Because of the generality of quantum graphs, such a shared viewpoint has also been found in wave neural networks using integrated photonic platforms [Nat. Photon. 11, 441 (2017): ref. 19] or diffractive layers [Science 361, 10041008 (2018): ref. 20].** To clarify this point, I included the discussion with related references in the revised manuscript.

(Lines 38-42) The use of network science is widespread throughout physics, as shown in the network modelling of material states³, potential landscapes⁸, and interacting quantum processors¹⁰, and in the field of quantum graph theory¹³⁻¹⁶.

(Lines 63-64) ... will provide an insight into complex materials and artificial neural networks in photonics, acoustics, quantum graphs, and other wave mechanics.

In the revised manuscript, I emphasized the following aspects of the novelty of my work against quantum graphs.

I. Extracting the kernel network from scattering

A quantum graph is built with (i) the metric graph quantifying the inter-particle distances, (ii) the Hamiltonian characterizing the physically defined graph edges with the metric graph, and (iii) the graph vertices reproducing the boundary condition between waves. Using quantum graphs, scattering phenomena are modelled by the scattering matrix that connects the incident and scattered fields (at the graph leads) through the internal waves (at the graph edges) where the internal waves are defined by the directions of graph edges and differential wave equations [PRL 85, 968 (2000): ref. 15]. **This method therefore corresponds to the network-based modelling of rigorous scattering matrix methods, fully describing the interactions between wave nodes.**

Although such a description allows for extracting important features in complicated systems composed of *a few particles*, **the rigorous description of conventional quantum graph theory, at the same time, hinders the extraction of the kernel part of wave networks**, especially when participating elements are numerous: e.g., examining scattering from disordered materials. **Therefore, when using quantum graph models, it is not straightforward to obtain the simplified and systematic network**

interpretation of scattering from disordered materials while maintaining physical validity, which is one of the goals of my work.

The solution for this challenging issue is to utilize the traditional approach in scattering problems: using the recursive form of the Lippman-Schwinger equation and the following Born series, and its approximation, which is applied to evolving scattering network model in the form of the firstorder Born approximation. As described in Supplementary Notes S1 and S4, I start from the structure factor originating from the approximation of the Lippman-Schwinger equation—the widely-used *integral* form of the governing equation to simplify scattering phenomena. Although the LippmanSchwinger equation and the differential wave equation are mathematically identical, **the recursive form of the Lippman-Schwinger equation enables the systematic approximation of scattering waves, allowing for extracting the network kernel in my work.**

In the revised manuscript, I clarified such a novelty of my work against quantum graph theory. As Reviewer 1 also pointed out a similar issue—the restriction of my work in describing general wave phenomena—I changed the terminology “wave network” to “scattering network”.

(Lines 48-53) Notably, quantum graphs provide an analytical tool to fully describe the interactions between wave nodes¹³⁻¹⁶. However, extracting the kernel of a wave network requires platform-specific simplification, as shown in the reflectionless and single-channel assumptions for the network modelling of guided^{19,27,33} and diffractive²⁰ systems. In this context, the efficient network modelling of scattering phenomena having complex interferences from multiple particles is still an open question.

(Lines 147-153) For example, quantum graphs enable the network modelling of scattering through the graph edges defined by the metric graph between particles and the governing Hamiltonian, and the graph vertices for field boundary conditions¹³⁻¹⁶. Although the quantum graph model for scattering corresponds to the network interpretation of a rigorous scattering matrix, this rigorous modelling, at the same time, hinders the extraction of the kernel part of wave networks, especially when participating elements are numerous, such as scattering from disordered materials.

(Lines 157-159) From the analysis of wave scattering using the Lippmann-Schwinger equation that allows for extracting kernel parts of scattering with the Born series⁵¹, I develop an evolution model for ...

II. Introducing the evolution process in network-based material design

As the reviewer commented, the key novelty of my manuscript is the “evolution” of a scattering network, which has not been studied in quantum graphs. **In this work, I introduced the concept of**

evolving scattering networks, which is fundamentally different from previous wave networks or quantum graphs in terms of handling “*open systems*”: allowing for the alteration of matter (*i.e.*, an increasing particle number) and energy (*i.e.*, minimizing the cost function) inside the system. **There exist only a few examples of handling time-varying quantum graphs, and these works are not about the evolving network structures with varying network sizes or topology but about time-varying unitary quantum evolution [Наносистемы: физика, химия, математика 6, 173 (2015)], which may correspond to generative network models.** I clarified this point in the revised manuscript.

(Lines 154-156) I also note that all the previous network structures applied to wave phenomena^{19-21,27,33}, including quantum graphs¹³⁻¹⁶, have employed static or generative models with time-independent network ...

2. Evolving networks is the key concept, but it is never defined.

Thank you very much for your comment. I included the definition of evolving network models:

(Lines 23-24) Evolving network models—the models that characterize the mechanisms and natures of timevarying networks—have stimulated significant advances in network science and related disciplines.

I also defined evolving scattering networks more rigorously to clarify the novelty of my work against previous efforts.

(Lines 65-67) Here, I propose the concept of evolving scattering networks—the open-system wave network models with a dynamically changing number of particles inside a system—which provides a novel tool for multiscale material design with target scattering responses.

3. what is the difference of this network method and just a traditional scattering method? Which network property is used here, eg in fig 3?

Compared with a traditional scattering method (e.g. collective coordinate approach) or evolutionary algorithms, the critical difference is the network concept based on wave interferences. The impact of the network interpretation is to decompose complex phenomena into simple sub-phenomena systematically. Although traditional approaches have been studied for designing wave scattering, these methods include neither the network viewpoint nor open-system natures—dynamically changing the system size (as stated in the reply to Point 1, quantum graphs have not been employed to evolving networks). **Therefore, it is difficult to examine the impact of an individual network element in the traditional methods, lacking the bridge between scattering phenomena and wave neural networks that require individual elements, i.e. “neurons”.**

In this resubmission, I revised the manuscript to clarify the above discussion.

(Lines 534-537) Because all of the existing wave neural networks have treated static networks¹⁷⁻²³, and traditional evolutionary algorithms lack the network model for characterizing interacting neurons^{34,35}, the concept of evolving scattering networks inspires the realization of neuroevolution¹² in wave physics.

In this context, Fig. 3d-l demonstrates the length-scale-dependent network properties through the node degree distributions. As the reviewer stated, the nature of SHU materials—crystal-like longrange order and Poisson-like short-range order—is well known; it may be textbook results. **However, its quantification using network parameters—node degrees—is a new result and demonstrates such an intermediate property using the network analysis for the first time.** Also, the visualization of node degrees (Fig. 3j-l) not only shows the statistical homogeneity of hyperuniformity but also presents a sharp distinction when compared with the evolving network results in Fig. 4a-f and 5a,d,g. Therefore, I'd like to maintain the current form of Fig. 3, while clarifying that the results themselves have been well-known.

(Lines 289-293) With this reciprocal-space design process, I revisit the comparison among the uncorrelated Poisson disorder (Fig. 3a), evolving SHU material (Fig. 3b), and ... to interpret the length-scale natures of each material state with the network concept.

4. can the proposed method outperform current ones?

The optimization time consumption and computing resources are not the topic of this manuscript. **However, the advantage of the proposed method is clear when handling dynamically-varying scattering systems, as shown in Fig. 4—the first demonstration of material screening with SHU.**

The difference of my approach compared to previous ones corresponds to the difference between evolving networks (changing node number) and static networks (fixed node number), or more generally, the difference between “open” systems and “closed” systems. **The proposed method is in sharp contrast to the conventional energy minimization with a fixed number of particles, which satisfies the definition of “closed” systems.**

Therefore, for open-system problems, such as the alteration of a given system with the additional inclusion of elements (e.g., Fig. 4), the evolution process is a more proper and natural strategy, as shown in the SHU conservation with evolving network models in Fig. 4. Although the final states of a material (or a network) obtained from those two approaches may not always be different, as analogous to the deterministic realization of scale-free networks [Physica A, 299, 559 (2001)], **the application of**

the evolution process can reveal unexplored system states, as shown in the seminal finding of scale-free networks [Science 286, 509 (1999)], and the discovery of super-dense material phases in short-range order in Figs 5 and 6 by introducing the concept of “preference” for material design for the first time.

Reply to Reviewer 3's report

In this paper, the author developed a novel wave-network scheme for the design and optimization of photonic network materials, which was demonstrated in stealthy hyperuniform material systems. In particular, the author formulated the material optimization problem, in which the materials are composed of a varying number of isotropic scatters, into an evolving wave network in the context of scattering. **One key novelty of the work is the proposed mapping between network science and wave physics, through the definition of nodes, weighted links, degree distributions, and evolution processes based on scattering theory.** The utility of the framework was demonstrated by designing and achieving extraordinary material states such as effectively denser or sparser particle distributions for short-range order while preserving long-range order of crystals or Poisson processes. **This approach provides the network-based interpretation of wave phenomena, extending the candidate platforms for wave neural networks.**

The paper is overall very well written and of great topical interest. The mapping between network and wave physics, as well as the concept using evolving network for photonic material design are very novel and inspiring. A key advantage compared to previously work on photonic material optimization and hyperuniform material design is that the current work allows varying particle numbers, enabling multi-scale open material system design. **I'm happy to recommend its publication once the author addresses the following comments and suggestions for minor revisions:** Thank you very much for providing the careful review and encouraging comments on my work.

Following the reviewer's comments and suggestions, especially the comparison with the collective coordinate approach, the quality of my manuscript is greatly improved.

Although viewing material scattering as a wave network is very inspiring, the key quantity of interest that characterizes the scattering behavior is the statistic structure factor, which was the objective function employed in the optimization. In this regard, the current approach, especially on the fundamental mathematical level, bears similarity with the collective coordinate approach developed by Torquato and co-workers, which were discussed in many of the references cited in the paper. **The key differences between the two include: (i) the collective coordinate approach does not explicitly consider an underlying "network" for the wave physics (ii) it does not allow varying particle numbers in its current implementation.** It might be useful for the author to elaborate these points a little in the paper.

I sincerely appreciate the reviewer for this insightful comment, clarifying the novelty of my work when compared with the collective coordinate approach. Following the reviewer's suggestion, I included the extended discussion for this point in the revised manuscript.

(Lines 340-344) Compared with the evolving scattering network model, the collective coordinate method does not explicitly consider an underlying network structure for wave physics. Furthermore, because the potential energy in the collective coordinate method is defined for the system of a *fixed* number of particles, the method does not allow varying particle numbers in its current implementation, as similar to ...

Some minor points:

P3, L65, "...suggested network modeling..." should be "... suggested network model..."?

Thank you very much for this careful comment. I have corrected this part in the revised manuscript.

In Fig 1, the network modeling for scattering process was not very apparent, as the links were not shown?

I appreciate the reviewer's helpful suggestion. To emphasize the network modelling, I included the links in the revised Fig. 1f, while I assume the plotting of a part of link weights (significant values of $|w_{p,q}^{\mathbf{k}}|$) because the network is fully connected.

(Lines 133-139) Red and blue solid lines represent the positive and negative signs of existing link weights defined by Eq. (5), respectively. Red and blue arrows also represent the positive and negative signs of newly included link weights after adding the $(n+1)^{\text{th}}$ particle, respectively. Only the links with significant values of $|w_{p,q}^k|$ are assumed to be plotted because a scattering network is fully-connected. The black arrow describes the \mathbf{k} -impulse component $\cos[\mathbf{k} \cdot (\mathbf{r}_p - \mathbf{r}_q)]$ of the link weight between the p^{th} and q^{th} particles. The transparency of the solid lines and arrows denotes the magnitude of the weights.

Decision Letter, first revision:

Dear Dr. Yu,

Thank you for submitting your revised manuscript "Evolving scattering networks for engineering disorder" (NATCOMPUTSCI-22-1027A). It has now been seen by the original referees and their comments are below. The reviewers find that the paper has improved in revision, and therefore we'll be happy in principle to publish it in Nature Computational Science, pending minor revisions to satisfy

the referees' final requests and to comply with our editorial and formatting guidelines.

TRANSPARENT PEER REVIEW

Nature Computational Science offers a transparent peer review option for original research manuscripts. We encourage increased transparency in peer review by publishing the reviewer comments, author rebuttal letters and editorial decision letters if the authors agree. Such peer review material is made available as a supplementary peer review file. **Please state in the cover letter 'I wish to participate in transparent peer review' if you want to opt in, or 'I do not wish to participate in transparent peer review' if you don't.** Failure to state your preference will result in delays in accepting your manuscript for publication.

Thank you again for your interest in Nature Computational Science Please do not hesitate to contact me if you have any questions.

Best,
Fernando

--

Fernando Chirigati, PhD
Chief Editor, Nature Computational Science
Nature Portfolio

ORCID

Reviewer #1 (Remarks to the Author):

Dear Editor,

I consider that the author has address in full my comments and I am satisfied with their response.

Consequently, I am happy to recommend the manuscript for publication.

Reviewer #1 (Remarks on code availability):

The author has included the codes necessary for reproducing Figs. 4 and 5. I consider these to be the most relevant in the manuscript. Personally I considered there is no need to include codes for reproducing Figs. 2 and 3, but others may disagree.

Reviewer #2 (Remarks to the Author):

I appreciate that the revise manuscript is much improved, both for clarity, setting against other methods and explanation of its novelty.

Therefore I am happy to change my mind and recommend it for publication in Nature Computational Science.

Reviewer #3 (Remarks to the Author):

In this revised version, the authors have constructively addressed the comments from all reviewers. The new table that includes explicit comparison with existing methods was very helpful. It can be accepted in the current form.

Author Rebuttal, second revision:

Reply to Reviewer 1's report

Dear Editor,

I consider that the author has address in full my comments and I am satified with their response.

Consequently, I am happy to recommend the manuscript for publication.

Reviewer #1 (Remarks on code availability):

The author has included the codes necessary for reproducing Figs. 4 and 5. I consider these to be the most relevant in the manuscript. Personally I considered there is no need to include codes for reproducing Figs. 2 and 3, but others may disagree.

I sincerely appreciate the reviewer's recommendation. In the revision, I also included the codes for reproducing Fig. 2 and 3 through the public repository. Thank you very much for your efforts during the peer review process.

Reply to Reviewer 2's report

I appreciate that the revise manuscript is much improved, both for clarity, setting against other methods and explanation of its novelty.

Therefore I am happy to change my mind and recommend it for publication in Nature Computational Science.

I sincerely appreciate the reviewer's recommendation. As I stated in the previous review, the comment on quantum graph theory was undoubtedly helpful in demonstrating the novelty of my work. Thank you very much for your efforts during the peer review process.

Reply to Reviewer 3's report

In this revised version, the authors have constructively addressed the comments from all reviewers. The new table that includes explicit comparison with existing methods was very helpful. It can be accepted in the current form.

I sincerely appreciate the reviewer's recommendation. Thank you very much for confirming the revised manuscript and your significant efforts during the peer review process.

Final Decision Letter:

Dear Professor Yu,

We are pleased to inform you that your Article "Evolving scattering networks for engineering disorder" has now been accepted for publication in Nature Computational Science.

Once your manuscript is typeset, you will receive an email with a link to choose the appropriate publishing options for your paper and our Author Services team will be in touch regarding any additional information that may be required.

Please note that *Nature Computational Science* is a Transformative Journal (TJ). Authors may publish their research with us through the traditional subscription access route or make their paper immediately open access through payment of an article-processing charge (APC). Authors will not be required to make a final decision about access to their article until it has been accepted. [Find out more about Transformative Journals](https://www.springernature.com/gp/open-research/transformative-journals)

Authors may need to take specific actions to achieve [compliance with funder and institutional open access mandates](https://www.springernature.com/gp/open-research/funding/policy-compliance-faqs). If your research is supported by a funder that requires immediate open access (e.g. according to [Plan S principles](https://www.springernature.com/gp/open-research/plan-s-compliance)) then you should select the gold OA route, and we will direct you to the compliant route where possible. For authors selecting the subscription publication route, the journal's standard licensing terms will need to be accepted, including [self-archiving policies](https://www.springernature.com/gp/open-research/policies/journal-policies). Those licensing terms will supersede any other terms that the author or any third party may assert apply to any version of the manuscript.

Acceptance of your manuscript is conditional on all authors' agreement with our publication policies (see <https://www.nature.com/natcomputsci/for-authors>). In particular your manuscript must not be published elsewhere and there must be no announcement of the work to any media outlet until the publication date (the day on which it is uploaded onto our web site).

Before your manuscript is typeset, we will edit the text to ensure it is intelligible to our wide readership and conforms to house style. We look particularly carefully at the titles of all papers to ensure that they are relatively brief and understandable.

Once your manuscript is typeset and you have completed the appropriate grant of rights, you will receive a link to your electronic proof via email with a request to make any corrections within 48 hours. If, when you receive your proof, you cannot meet this deadline, please inform us at rjsproduction@springernature.com immediately.

If you have queries at any point during the production process then please contact the production team at rjsproduction@springernature.com. Once your paper has been scheduled for online publication, the Nature press office will be in touch to confirm the details.

Content is published online weekly on Mondays and Thursdays, and the embargo is set at 16:00 London time (GMT)/11:00 am US Eastern time (EST) on the day of publication. If you need to know the exact publication date or when the news embargo will be lifted, please contact our press office after you have submitted your proof corrections. Now is the time to inform your Public Relations or Press Office about your paper, as they might be interested in promoting its publication. This will allow them time to prepare an accurate and satisfactory press release. Include your manuscript tracking number NATCOMPUTSCI-22-1027B and the name of the journal, which they will need when they contact our office.

About one week before your paper is published online, we shall be distributing a press release to news organizations worldwide, which may include details of your work. We are happy for your institution or funding agency to prepare its own press release, but it must mention the embargo date and Nature Computational Science. Our Press Office will contact you closer to the time of publication, but if you or your Press Office have any inquiries in the meantime, please contact press@nature.com.

We welcome the submission of potential cover material (including a short caption of around 40 words) related to your manuscript; suggestions should be sent to Nature Computational Science as electronic files (the image should be 300 dpi at 210 x 297 mm in either TIFF or JPEG format). We also welcome suggestions for the Hero Image, which appears at the top of our [home page](http://www.nature.com/natcomputsci); these should be 72 dpi at 1400 x 400 pixels in JPEG format. Please note that such pictures should be selected more for their aesthetic appeal than for their scientific content, and that colour images work better than black and white or grayscale images. Please do not try to design a cover with the Nature Computational Science logo etc., and please do not submit composites of images related to your work. I am sure you will understand that we cannot make any promise as to whether any of your suggestions might be

selected for the cover of the journal.

Best,
Fernando

--

Fernando Chirigati, PhD
Chief Editor, Nature Computational Science
Nature Portfolio

P.S. Click on the following link if you would like to recommend Nature Computational Science to your librarian: https://www.springernature.com/gp/librarians/recommend-to-your-library

** Visit the Springer Nature Editorial and Publishing website at www.springernature.com/editorial-and-publishing-jobs for more information about our career opportunities. If you have any questions please click here. **